# Epigenome and transcriptome changes in *KMT2D*-related Kabuki syndrome Type 1 iPSCs, neuronal progenitors and cortical neurons

Sara Cuvertino[1,2☯]*, Evgenii Martirosian[1,3☯], Kedar Bhosale[1,2], Peiwen Cheng[1,2], Terence Garner[3], Ian J. Donaldson[4], Adam Jackson[1,5], Adam Stevens[3], Andrew D. Sharrocks[6], Susan J. Kimber[2☯]*, Siddharth Banka[1,5☯]*

1 Division of Evolution, Infection and Genomics, School of Biological Sciences, Faculty of Biology, Medicine, and Health, The University of Manchester, Manchester, United Kingdom, 2 Division of Cell Matrix Biology and Regenerative Medicine, School of Biological Sciences, Faculty of Biology, Medicine, and Health, The University of Manchester, Manchester, United Kingdom, 3 Division of Developmental Biology & Medicine, School of Biological Sciences, Faculty of Biology, Medicine, and Health, The University of Manchester, Manchester, United Kingdom, 4 Bioinformatics Core Facility, The University of Manchester, Manchester, United Kingdom, 5 Manchester Centre for Genomic Medicine, St. Mary's Hospital, Manchester University Foundation NHS Trust, Health Innovation Manchester, Manchester, United Kingdom, 6 Division of Molecular and Cellular Function, School of Biological Sciences, Faculty of Biology, Medicine and Health, The University of Manchester, Manchester, United Kingdom

☯ These authors contributed equally to this work.
* Sara.Cuvertino@manchester.ac.uk (SC); Sue.Kimber@manchester.ac.uk (SJK); Siddharth.Banka@manchester.ac.uk (SB)

## Abstract

Kabuki syndrome type 1 (KS1) is a neurodevelopmental disorder caused by loss-of-function variants in *KMT2D* which encodes a H3K4 methyltransferase. The mechanisms underlying neurodevelopmental problems in KS1 are still largely unknown. Here, we track the epigenome and transcriptome across three stages of neuronal differentiation using patient-derived induced pluripotent stem cells (iPSCs) to gain insights into the disease mechanism of KS1. In KS1 iPSCs we detected significantly lower levels of functional *KMT2D* transcript and KMT2D protein, and lower global H3K4me1, H3K4me2 levels and modest reduction in H3K4me3. We identify loss of thousands of H3K4me1 peaks in iPSCs, neuronal progenitors (NPs) and early cortical neurons (CNs) in KS1. We show that the number of lost peaks increase as differentiation progresses. We also identify hundreds of differentially expressed genes (DEGs) in iPSCs, NPs and CNs in KS1. In contrast with the epigenomic changes, the number of DEGs decrease as differentiation progresses. Our analysis reveals significant enrichment of differentially downregulated genes in areas containing putative enhancer regions with H3K4me1 loss. We also identify a set of distinct transcription factor binding sites in differentially methylated regions and a set of DEGs related to KS1 phenotypes. We find that genes regulated by SUZ12, a subunit of Polycomb Repressive complex 2, are over-represented in KS1 DEGs at early stages of differentiation. In conclusion, we present a disease-relevant human cellular model for

**Data availability statement:** The data that support the findings of this study are publicly available from NCBI GEO with the identifier(s) GSE289158 and GSE289159.

**Funding:** We acknowledge the support of Great Ormond Street Hospital Charity (V4621 to SB, SK, AS, AS, SC), Newlife Charity (16-17/10 to SB, SK, AS, SC), Manchester University Hospitals NHS Foundation Trust Kabuki Research Fund (629396 to SB), the NIHR Manchester Biomedical Research Centre (NIHR203308 to SB), the MRC Epigenomics of Rare Diseases Node (MR/Y008170/1 to SB, SK) and the UKRI Engineering & Physical Sciences Research Council (EPSRC, EP/X027007/1 to KB) which is part of the Chrom_Rare consortium (HORIZON-MSCA-2021-DN-01-01). The funders had no role in study design, data collection and analysis, decision to publish, or preparation of the manuscript.

**Competing interests:** The authors have declared that no competing interests exist.

KS1 that provides mechanistic insights for the disorder and could be used for high throughput drug screening for KS1.

## Author summary

Kabuki syndrome is a genetic condition that affects multiple organ system and results in significant neurodevelopmental issues in a large majority of affected individuals. Kabuki syndrome, in most people, is caused by changes in a gene called *KMT2D* but we do not understand how these genetic changes result in neurodevelopmental issues. We performed experiments on stem cells generated from individuals affected with Kabuki syndrome to understand the effects of their genetic changes on development of brain cells. We find that Kabuki syndrome cells demonstrate a profound difference in the regulation and activity of hundreds of other genes in comparison with the control cells. Many of these genes have well established role in development and function of brain cells. These differences are present at all stages of the development, from stem cells to neuronal cells. In conclusion, these findings provide an insight into the mechanisms underlying Kabuki syndrome and potentially provide a model to test possible therapies.

## Introduction

*KMT2D* encodes a trithorax-related histone 3 lysine 4 (H3K4) methyltransferase, which is part of the COMPASS complex that primarily binds to enhancer regions and maintains global levels of H3K4me1 [1,2]. Mouse *Kmt2d* knockouts are embryonic lethal [3] and it is required for exit of mouse embryonic stem cells (ESCs) from the naive pluripotent state, but is dispensable for maintenance of cell-identity [4,5]. KMT2D is also crucial for the development of several organs including the brain [6,7].

Rare mono-allelic *KMT2D* variants cause Kabuki syndrome (KS) Type 1 (KS1, OMIM#147920) [8,9]. KS type 2 is caused by rare hetero- or hemizygous variants in a H3K27 demethylase encoding *KDM6A* gene (OMIM#300867) [10,11]. KS1 is characterised by distinct facial dysmorphism, mild to moderate intellectual disability, developmental delay, a range of internal organ malformations (congenital heart defects, skeletal defects, cleft palate and genitourinary malformations) and systemic problems (endocrine disorders, deafness, immune defects) [12].

Animal and cellular models for KS1 have provided valuable insights into the disease mechanisms [7,13,14]. *Kmt2d* zebrafish morphants show reduced hindbrain and midbrain size [6], and learning and memory impairment was reported in a *Kmt2d*$^{+/\beta Geo}$ murine model [7]. Suppression of oxygen-responsive gene expression has been seen in induced pluripotent stem cells (iPSCs) derived from patients with nonsense *KMT2D* variants [15]. Changes in locus-specific loss of gene expression were reported in KS1 iPSC models with truncating *KMT2D* variants [16]. Of note,

most studies have used models with null variants, even though ~10% KS1-causing variants are missense [17]. Furthermore, most studies have used single cell types. Thus our knowledge about the impact of KS1-causing *KMT2D* variants at different stages of differentiation is lacking.

Here, we track the epigenomic and transcriptional panorama across three stages of neuronal differentiation using patient-derived iPSCs to gain insights into disease mechanism of KS1. We show that in iPSCs, KS1 causing variants result in loss of KMT2D function and alter the H3K4me1 and transcriptional landscapes in a correlated manner. We show that epigenomic differences between KS1 and control cells increase with differentiation progression, but the transcriptional changes become less abundant. Our analysis also reveals a set of potential transcriptional regulators and their *cis*-regulatory elements, differentially expressed genes (DEGs) and master regulators that likely underpin the KS1 phenotypes.

## Results

### KS1 causing variants result in loss of *KMT2D* function in iPSCs

IPSCs were generated either from fibroblasts in the HipSci program (www.hipsci.org/#/cells) or in-house from fibroblasts or peripheral blood mononuclear cells and used three controls and three individuals with molecularly confirmed KS1. Our cohort included one individual each with frameshift (NM_003482.4: *KMT2D*; c.5527dupA, p. (Pro1849Ter)), missense (c.16019G>A, p. (Arg5340Gln)), and nonsense (c.16180G>T, p. (Glu5394Ter)) variants (Fig 1A and S1 Table).

Firstly, to investigate the effect of KS1 variants at the pluripotent stem cell stage, we accessed iPSC RNAseq datasets of 53 control lines and 7 KS1 lines (4 replicates with the missense variant, 2 with the nonsense variant and one with the frameshift variant) generated by the HipSci consortium and our lab (S1 Fig and S1 and S2 Tables). Most disease causing *KMT2D* variants are predicted to result in haploinsufficiency but it has not been proven to the best of our knowledge [8]. We, therefore, assessed the level of *KMT2D* mRNA using the merged RNAseq data. We did not observe significant difference in the total *KMT2D* mRNA level between controls and KS1 iPSCs (total *KMT2D* reads) (Figs 1B and S2A). However, quantification of reference and mutant *KMT2D* transcripts revealed that the reference *KMT2D* reads (REF) were significantly lower in all KS1 iPSCs compared to controls (Figs 1B, S2B – S2C). Interestingly, level of reference *KMT2D* reads (referred to as REF in Fig 1B) in KS1 iPSCs was higher compared to reads with mutations (referred to as ALT in Fig 1B). Furthermore, the level of reads with missense variants was significantly higher compared to the reads with truncating variants, likely due to nonsense-mediated mRNA decay in transcripts with premature truncating codons (Fig 1B).

Next, we performed Western blotting on nuclear protein extracts from iPSCs, which revealed a significant decrease in KMT2D protein level in nuclei of all KS1 iPSCs, including the ones with the missense variant, compared to control iPSCs (Figs 1C and S3A). As KMT2D is a H3K4 methyltransferase, we quantified the total levels of H3K4me1, H3K4me2, and H3K4me3 histone marks in all three KS1 and control iPSCs by Western blotting. In all three lines, including in the one with the missense variant, we observed a significant decrease in the levels of H3K4me1 and H3K4me2 histone marks with slightly reduced H3K4me3 histone mark (Figs 1D and S3B – S3D). Next, we examined the DepMap portal (https://depmap.org/portal/) for the levels of 42 different histone marks in cancer cell lines with *KMT2D* mutations and compared to cell lines without *KMT2D* damaging mutations (S4 Fig). We found H3K4me1 and H2K4me2 levels to be most significantly reduced in cell lines with *KMT2D* mutations compared to cell lines without *KMT2D* mutations.

Next, we evaluated if loss of KMT2D function led to any changes in the molecular pluripotency state of KS1 iPSCs. In the merged RNAseq dataset, we did not find significant differences in the expression of key genes related to pluripotency in KS1 iPSCs compared to control iPSCs (Fig 1E). We validated these results by RTq-PCR and immunofluorescence staining for SOX2, NANOG and OCT4 (S5 Fig).

Overall, these data show that KS1-causing heterozygous *KMT2D* variants, including missense variants, are associated with loss of function that is reflected in significant decrease in levels of H3K4me1/me2. The loss of function resulting from KS1-causing heterozygous *KMT2D* variants does not result in major changes in pluripotency-associated gene expression.

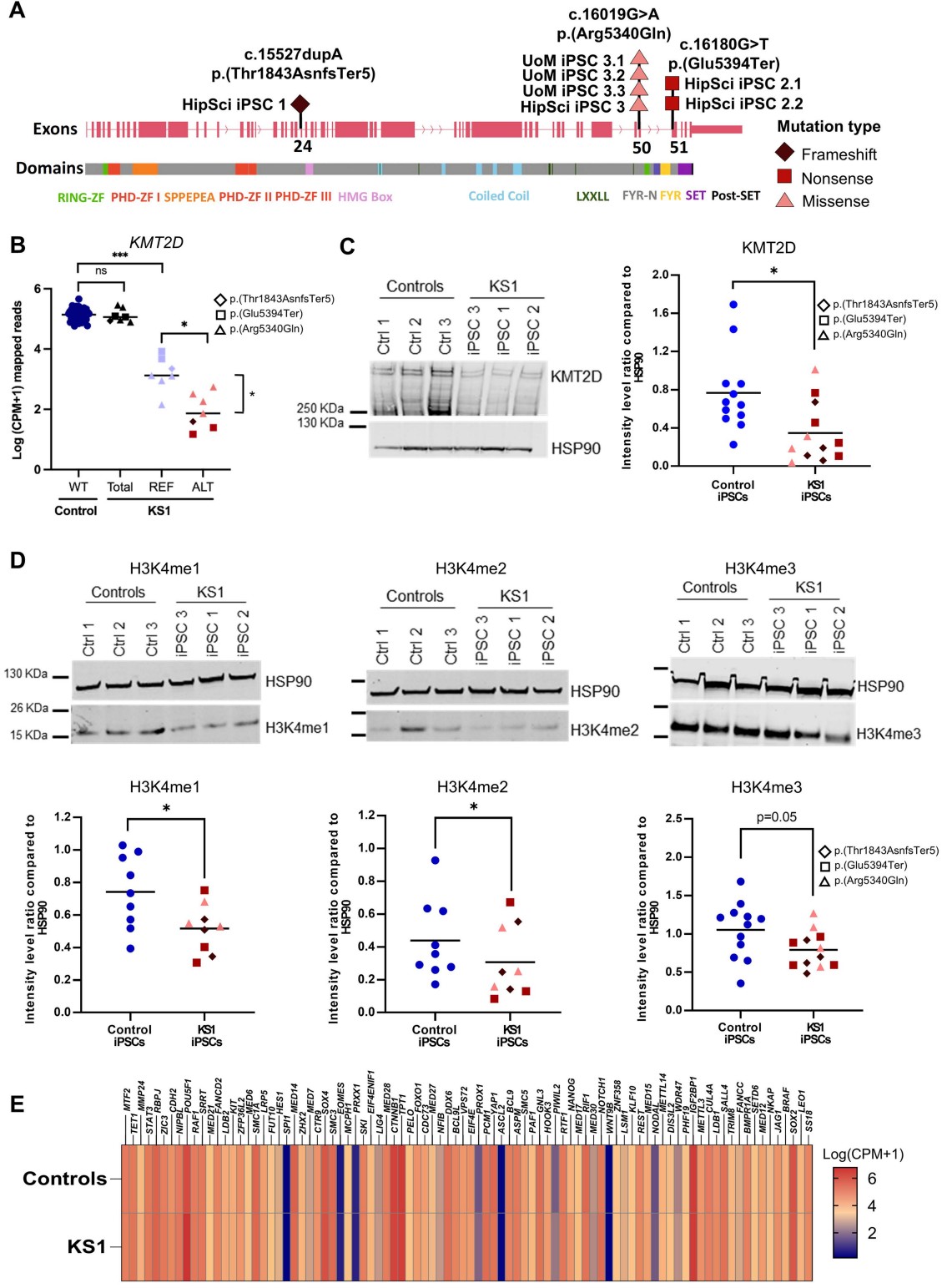

**Fig 1. Analysis of KMT2D expression in KS1 iPSCs. A)** Schematic representation of *KMT2D* exons and protein domains and regions with the differ-ent *KMT2D* variants highlighted in pink, red and brown for missense, nonsense and frameshift mutations, respectively. (Dataset 1: DS1-UoM; Dataset 2: DS2-HipSci). **B)** Log counts per million (CPM+1) mapped reads for *KMT2D* transcript detected in the integrated RNAseq analysis. REF shows reads

with reference sequence – no mutation (light blue) and ALT shows reads with mutation (pink/red/brown as in A). Unpaired T-test was performed to compare differences in number of transcripts in controls (n=53) and KS1 (n=7), difference in number of REF transcripts in controls and KS1 (***p<0.001), number of REF and ALT transcripts in KS1 (*p<0.05) and number of ALT transcripts between KS1 samples with missense mutations and KS1 samples with protein truncating mutations (frameshift and nonsense) (*p<0.05). **C)** Representative image for Western blot for KMT2D and scatter plot showing KMT2D quantification relative to loading control, HSP90 (n=4; * p<0.05). Variants shown in pink/red/brown as in A. **D)** Representative images for Western blots for H3K4me1 ($n = 3$), H3K4me2 ($n = 3$) and H3K4me3 ($n = 4$) and scatter plots showing their quantification relative to loading control, HSP90 (*p<0.05). Variants shown in pink/red/brown as in A. **E)** Log counts per million (CPM+1) mapped reads for pluripotency related transcripts detected in the integrated RNAseq analysis.

## H3K4me1 landscape is dysregulated in KS1 iPSCs

KMT2D is a major H3K4 mono-methyltransferase essential for enhancer activation during cell differentiation [1,3]. Since we observed a significant global decrease in H3K4me1 in our cell lines (Fig 1D), we decided to investigate the H3K4me1 landscape in KS1 in more detail. As limited studies have been conducted on KS1 animal or cellular models with missense variants, we performed H3K4me1 ChIPseq in one KS1 (c.16019G>A, p. (Arg5340GIn)) and one control iPSC line, both in triplicate. We detected 9,309 out of 205,973 peak sites to be significantly differentially methylated (FDR<0.05; -1>Log$_2$FC>1). Of these, 8,146 sites (87.5% of all differentially methylated peak sites) showed loss, and 1,163 sites showed gain in the H3K4me1 peak signal (Fig 2A - 2C). H3K4me1 peak site losses were enriched in cell-type independent EpiMap annotated enhancer regions (Fisher's test; p-value<2.2e$^{-16}$) (Figs 2C, S6A and S7).

To identify transcription factor binding sites predicted to be affected by changes in H3K4me1 levels, we performed motif analysis on significantly differentially methylated peak sites. This showed these sites to be associated with motifs predicted to be binding sites for Zinc Finger (ZNF) transcription factors family members, as well as the transcription factor WT1 (Fig 2D). Cell-type-independent enhancers with H3K4me1 loss were enriched for the WT1 transcription factor motif, while cell-type-independent promoters with H3K4me1 loss were associated with the EGR1 motif (S6B – S6C Fig). KS1 sites that gained H3K4me1 peaks showed significant enrichment for KLF1 transcription factor binding sites (Fig 2D).

Overall, these data show significant dysregulation of the H3K4me1 landscape in KS1 iPSCs, especially in the enhancer regions.

## The transcriptome is altered in KS1 iPSCs and is correlated with the H3K4me1 landscape

Altered H3K4me1 landscape is predicted to result in changes in the transcriptome. We, therefore, performed RNAseq on control iPSCs and KS1 iPSC harbouring missense mutation in *KMT2D* (c.16019G>A, p. (Arg5340Gin)) (S8A Fig) and detected 909 significant DEGs (FDR<0.05, -1>Log2FC>1). Of these, 554 DEGs were downregulated while 355 were upregulated (Fig 3A). Gene Ontology (GO) overrepresentation analysis on the 909 DEGs detected biological categories related to organ morphogenesis and development, cell adhesion and actin and extracellular structure to be significantly overrepresented in DEGs (FDR<0.05) (Fig 3B). Notably, DEGs related to stem cell differentiation and mesenchyme development (*TBX2, TWIST1, GBX2, MEF2C*), negative regulation of locomotion (*ANGPT2, GREM1*), morphogenesis of an epithelium (*AJAP1, GDF7*) and regulation of neuron projection development (*POU3F2, NTRK2*) were upregulated while actin filament-based movement (*TNNC2, ACTA1*) and sensory perception of light stimulus (*TULP1, AIPL1, CRYBB1*) were downregulated (Fig 3A). GO categories for ear development (*TSHZ1, BMP5*), neuron projection (*GDF7, CNTN6*), sensory organ morphogenesis (*DIO3, NTRK2*), embryonic organ development (*PAX5, KDR*) and heart morphogenesis (*SNAI2, PITX2*) included both up and downregulated DEGs (Fig 3B).

Next, we performed transcription regulator enrichment analysis on DEGs. This revealed the upregulated DEGs (adjusted p-value<0.05) being over-represented by regions of DNA associated with SUZ12 recruitment, a component of the Polycomb Repressive Complex 2 (PRC2) (S3 Table).

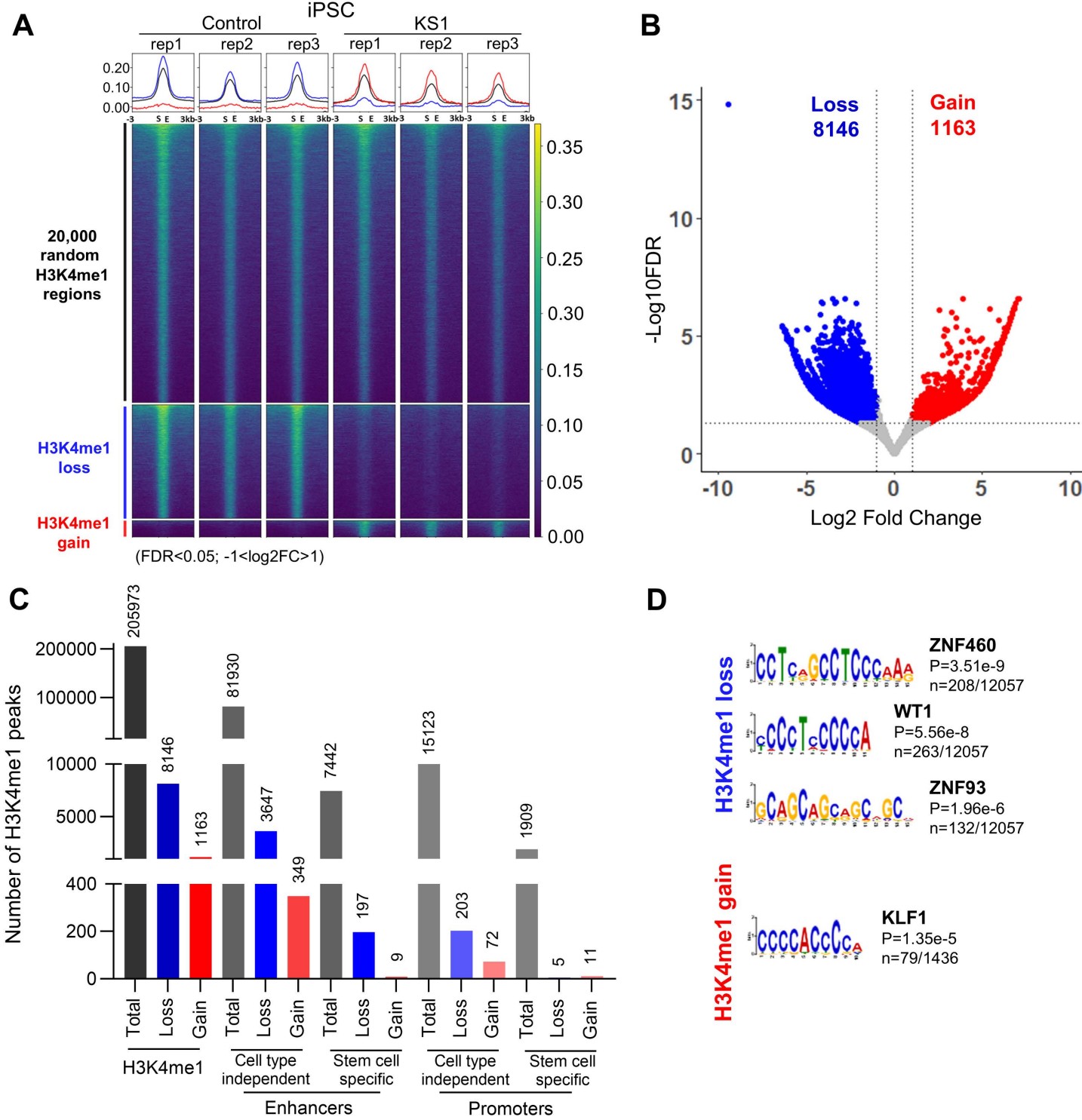

**Fig 2. Loss of H3K4me1 in KS1 iPSCs.** **A)** Heatmap of H3K4me1 peaks (subset of consensus regions 20,000 out of 205,973; black), H3K4me1 loss regions (n = 8146; blue), and H3K4me1 gain regions (n = 1163; red) in KS1 (iPSC 3) compared to control (Ctrl 1). **B)** Volcano plot showing loss (blue) and gain (red) in H3K4me1. **C)** Bar graph showing number of H3K4me1 peaks in relation to enhancer and promoter regions for total, loss and gain in mono-methylation. Cell-type independent enhancers and promoters were imputed by EpiMap from 833 biosamples and cell-type dependent enhancers and promoters were imputed by EpiMap from DF19.11 iPSCs. **D)** Top TF motifs predicted in differential H3K4me1 peaks (loss and gain of H3K4me1 in binding regions) between KS1 and control. Number (n) represents the number of times the motif was found within the unique sequences underlying the differential H3K4me1 regions.

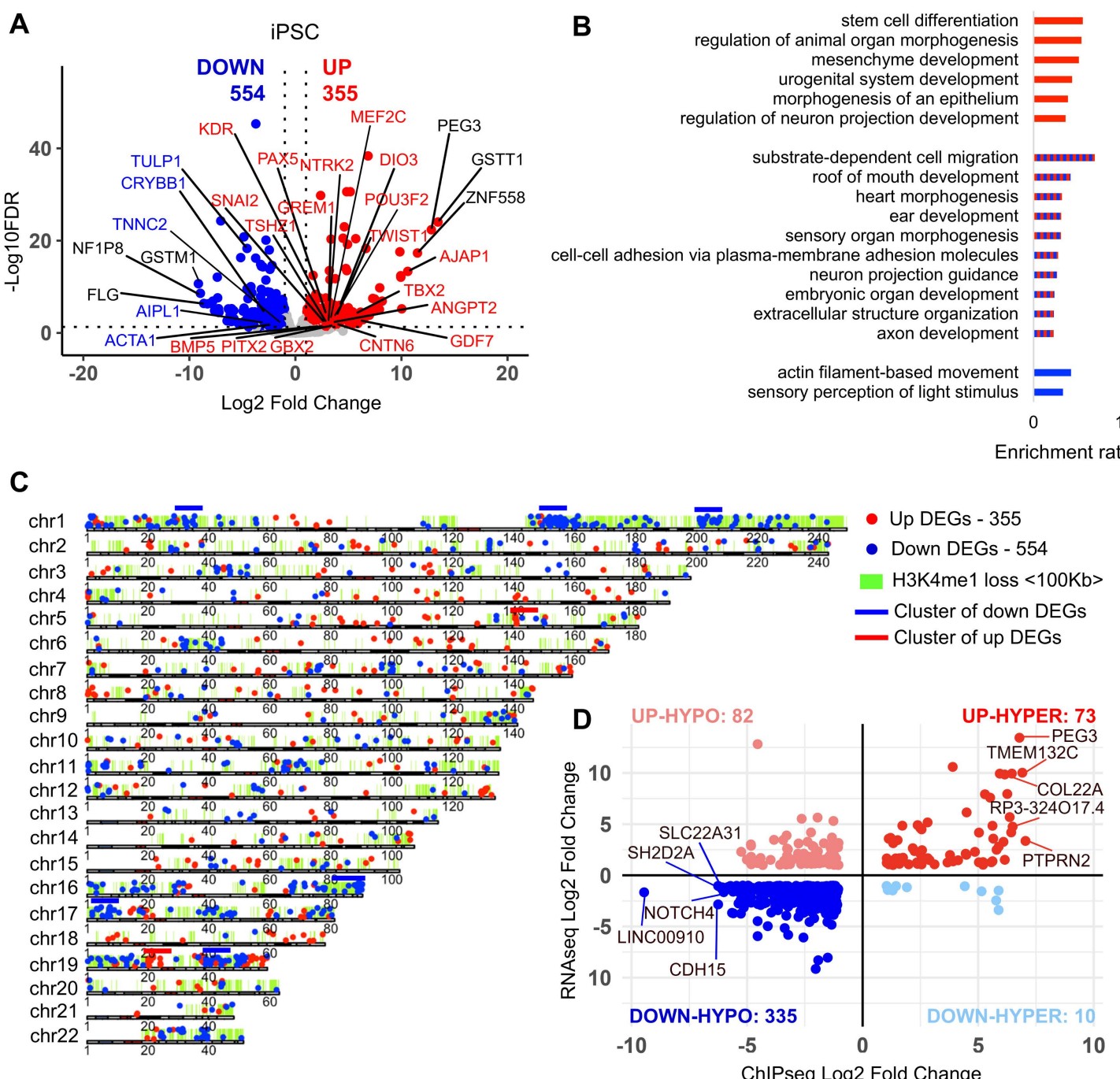

**Fig 3. Correlation between changes in transcripts and H3K4me1 in KS1 iPSCs. A)** Volcano plot showing down and up regulated genes on autosomal chromosomes in KS1 (iPSC 3) compared to controls (Ctrl 1) (adjp<0.05, -1>Log2FC>1). Genes highlighted are related to the GO categories in blue and red. Top 3 genes with highest or lowest Log2FC are highlighted by black text. **B)** WebGestalt was performed on DEGs (adjp<0.05; -1>Log2FC>1) between KS1 iPSC and control iPSC. Gene ontology categories for biological processes are shown in the bar graphs (FDR<0.05). Red bars correspond to up DEGs, blue to down DEGs and red and blue to both. **C)** Karyoplot showing up (red) and down (blue) DEGs on each chromosome. Lime-green regions represent 100Kb up and downstream regions within H3K4me1 loss. Different grey colours represent the different G band staining and red represents centromere. Blue lines above chromosomes represent regions of enrichment of downregulated DEGs and red lines for upregulated DEGs (FDR<0.05; hypergeometric test) **D)** Scatter plot showing change in expression and change in H3K4me1 within ±100kb of DEGs.

Downregulation of several genes within a 110 Kb region on chromosome 19 was previously reported in KS1 iPSCs suggesting locus-specific targeting of KMT2D [16]. We, therefore, asked if any clusters of DEGs were present in our data-set as well. We found significant enrichment of downregulated DEGs located on chromosome 1, 16, 17 and 19 (hyper-geometric test, FDR < 0.0001) (Figs 3C and S8C). We also detected a region on chromosome 19, which overlaps with the previously reported downregulated cluster at 19q13.33 [16].

Next, we asked if there was correlation between the altered H3K4me1 signals and gene expression in our cell model. For this, we analysed expression of genes within ±100kb of differentially lost H3K4me1 regions. We found several regions of the genome with clusters of reduced H3K4me1 peaks and downregulated genes with significant enrichment of downregulated DEGs on chromosome 1, 16, 17 (hypergeometric test, FDR < 0.0001) (Figs 3C and S8C). We also found H3K4me1 and gene expression levels to be significantly correlated based on the ± 100kb distance between gene TSS and H3K4me1 peak (p-value < 0.05; hypergeometric test) (Fig 3D and S4 Table). Most of these genes correspond to areas near cell-type independent enhancer regions with loss of H3K4me1 (S8D Fig).

Overall, these data show that at the stem cell stage there are significant changes in expression of genes related to the KS1 phenotype, with SUZ12 as top regulatory factor among the upregulated genes.

## The neuronal progenitor and neural epigenomes are altered in KS1

Having shown the epigenomic changes at the stem cells stage, we asked if similar changes could also be present in differentiating cells. Learning difficulties are seen in 95% individuals and epilepsy is present in 25% individuals affected by KS [9,18]. Such difficulties may arise because early development of neurons and neural progenitors is abnormal such as by generation of fewer or incompetent subsets of neurons. Additionally, neurodevelopmental disorders can be inferred by the impairment of neural progenitor cell population [19]. We, therefore, differentiated the KS1 iPSCs harbouring mis-sense mutation in *KMT2D* (c.16019G>A, p. (Arg5340Gln)) and control iPSCs into neuroepithelium with 10 days of SMAD signalling inhibition [20] giving rise to neural rosettes (S9A Fig) and later into early cortical neurons (Day 30) (Figs 4A and S10A). After the formation of the rosettes, cells were enriched for neural progenitors by FACs for a CD44⁻CD184⁺CD24⁺ population at Day 18 and for neuronal cells by sorting for a CD44⁻CD184⁻CD24⁺ population at Day 30 (Fig 4A). Both control and KS1 cells expressed key markers of neural progenitors such as *PAX6, FOXG1, SOX2, NES* and *OTX2*, and of neuronal cells such as *TBR1, MAP2* and *GRIA2* (Fig 4B).

We performed H3K4me1 ChIPseq on sorted neural progenitors and neuronal cells. In neural progenitors, we detected 22,975 differentially methylated sites (FDR < 0.05; -1 > Log2FC > 1). Of these, 15,452 (67.2%) sites showed loss and 7,523 showed gain in H3K4me1 in KS1 cells (Figs 4C, 4E and S10). In neurons, we detected 25,858 differentially methylated sites (FDR < 0.05; -1 > Log2FC > 1). Of these, 21,586 sites showed loss in H3K4me1 and 4,272 showed gain in H3K4me1 (Figs 4D, 4F and S11). Of the H3K4me1 peak site losses, 5,796 (37.5%) in neural progenitor and 8,321 (38.5%) in neu-rons correspond to cell-type independent enhancer regions (Figs 4E - 4F; S9B – S9C and S10B – S10C).

Motif analysis on neural progenitors revealed that the sites where H3K4me1 was lost in KS1 cells were predicted to be targets for ZNF family transcription factors as well as OTX, RARG and RFX5 while sites where H3K4me1 was gained were predicted to be targets for IKZF1, KLF4, FOXO6 and TFAP2C (Fig 4G). ZNF460, PATZ1, PITX1 and ZKSCAN1 were found to be associated with cell-type independent enhancers with H3K4me1 loss while ELF1, TFAP2C and KLF1 with cell-type independent enhancers with H3K4me1 gain (S9D Fig). In neuronal cells, loss of H3K4me1 sites were predicted to be bound by ZNF transcription factor family members, BHLHA15, KLF4, NFIC and ESRRA (Fig 4H). ZNF transcription factor family members, NEUROD1, ARGFX, NFIB and NFIC motifs were found to be associated with cell-type indepen-dent enhancers with H3K4me1 loss; ZNF was also found in cell-type independent promoters with H3K4me1 loss (S10D – S10E Fig).

Next, we compared the significantly differentially methylated peaks across the three stages of neuronal differentiation. We identified 88 H3K4me1 regions (74 losses and 14 gains) that were affected across all stages in the KS1 lines (Fig 4I).

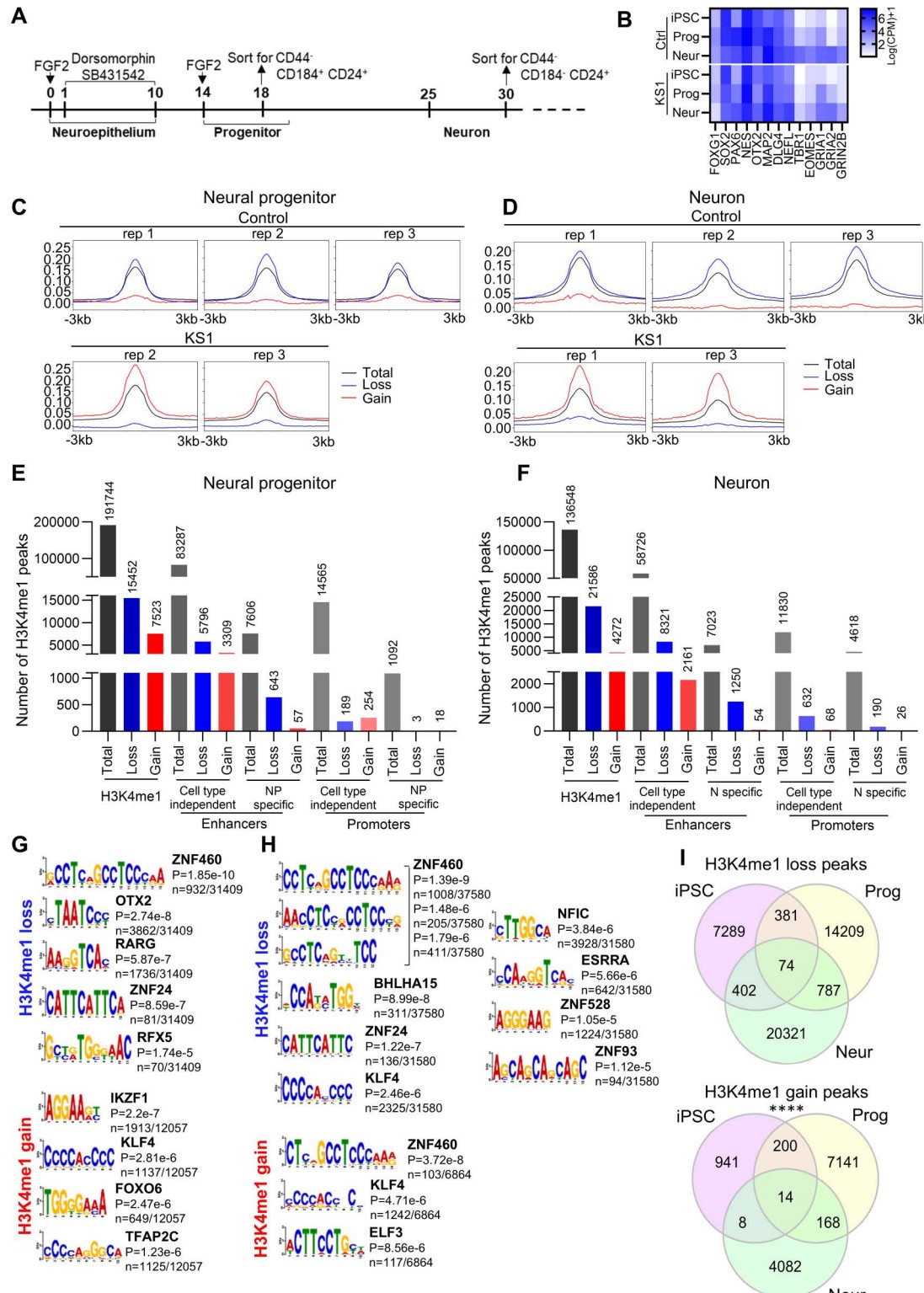

**Fig 4. Loss of H3K4me1 in KS1 neural progenitors and neurons. A)** Schematic representation of the neuronal differentiation protocol. **B)** RNAseq analysis showing transcript level for key neuronal markers in control (Ctrl 1) and KS1 (iPSC 3) samples at stem cell, progenitor and neuronal stages. **C-D)** Line graph of H3K4me1 peaks (black), H3K4me1 loss regions (blue), and H3K4me1 gain regions (red) for neuronal progenitors and neurons. **E-F)**

Bar graph showing number of H3K4me1 peaks in relation to enhancer and promoter regions for total, loss and gain in mono-methylation. Cell-type independent enhancers and promoters were imputed by EpiMap from 833 biosamples and cell-type dependent enhancers and promoters were imputed by EpiMap either from (**E**) H9-derived neural progenitors or from (**F**) H1-derived neurons. **G-H)** Top TF motifs predicted in differential H3K4me1 peaks (loss and gain of H3K4me1 in binding regions) between KS1 and control. Number (n) represents the number of times the motif was found within the unique sequences underlying the differential H3K4me1 regions. **I)** Intersection analysis showing H3K4me1 peaks in common between iPSC, Progenitor (Prog) and Neurons (Neur) (hypergeometric test; ****p<0.0001) between KS1 (c.16019G>A) and control.

Four hundred and fifty five H3K4me1 loss peaks were maintained between iPSC and neural progenitor stage and over 800 between neural progenitor and neuronal cells stage (hypergeometric test; p-value<0.0001) (Fig 4I).

Overall, in our KS1 model we witness increasing dysregulation of the H3K4me1 landscape with advancing differentiation from iPSCs to neurons.

## The transcriptome is altered in KS1 neural progenitors and neurons

Finally, we investigated the transcriptome of KS1 neural progenitors and neurons by RNAseq (S8B Fig). In neural progenitors, we detected 447 DEGs (FDR<0.05, -1<LogFC>1). Of these, 264 DEGs were downregulated while 183 DEGs were upregulated in KS1 neural progenitors (Fig 5A). GO analysis on the 447 DEGs revealed biological categories related to organ development, cell fate commitment and extracellular structure (FDR<0.05) (Fig 5C). Notably, DEGs related to neuron projection guidance (*FEZ1, VANGL2*) and cilium organization (*CDK10, DNALI1*) were downregulated whereas morphogenesis of epithelium (*AJAP1, IRX2*) and embryonic organ development (*TBX1, EN2*) were upregulated. GO categories for urogenital system development (*PAX8, AMH*), negative regulation of nervous system development (*INPP5F, SOX8*), and synapse organisation (*SLC7A11, NTNG2*) were contributed by both up- and downregulated DEGs. In neuronal cells, we detected a lower number of DEGs with only 116 DEGs (FDR<0.05, -1>Log2FC>1) and 147 DEGs (FDR<0.1, -1>Log2FC>1) (Fig 5B). Of these, 31 (44; FDR<0.1) DEGs were downregulated while 85 (103; FDR<0.1) DEGs were upregulated (Fig 5B). GO analysis on the 147 DEGs (FDR<0.1) showed biological categories related to nerve development (*TBX1, NGFR, EMX1*) contributed by both up and down DEGs while GO categories related to pattern specification process (*HOXB2, HOXB3, BARX1*), autonomous nervous system development (*SOX10, TBX1*) were upregulated (Fig 5D).

Transcription factor enrichment analysis revealed SUZ12 as top regulatory factor for upregulated DEGs in neural progenitors (adjusted p-value<0.05) (S3 Table). SUZ12 was also the top most hit for neurons, although not statistically significant. Additionally, we found significant enrichment of downregulated DEGs located on chromosome 1, 16 and 17 in neural progenitors and in 1, 5 and 16 in neuronal cells (hypergeometric test, p-value<0.0001) (S11 - S12 Figs).

Furthermore, we assessed positional correlation between altered H3K4me1 peaks and DEGs. In neural progenitors and neurons, we found statistically significant association between upregulated DEGs and H3K4me1 gain at distances 5kb to 100kb between the H3K4me1 peak and gene TSS (hypergeometric test; p-value<0.05) (S4 Table). Downregulated DEGs were associated with H3K4me1 loss peaks at 5kb to 25kb distances only in neural progenitors (S4 Table). Analysis of positional association of DEGs, revealed clusters of downregulated DEGs in neural progenitors at chromosomes 1, 2, 16 and 17 (hypergeometric test; FDR<0.05) (S13A Fig). In neuronal cells there was one cluster of upregulated DEGs on chromosome 1 (hypergeometric test; FDR<0.05) (S13B Fig).

Next, to analyse which changes in the transcriptome were in common at all three stages of neuronal differentiation, we compared DEGs between the three differentiation stages. Interestingly, we observed 50 significant DEGs (13 down- and 37 upregulated) in common across all three stages (Fig 5E). Intersection analysis showed most genes were significantly dysregulated between the iPSC and neural progenitor stage (exact test; p<0.0001).

To identify possible direct associations between H3K4me1 loss and downregulated genes across multiple stages of iPSC/differentiation in our model, we extracted genes, which are downregulated and have transcription start sites within

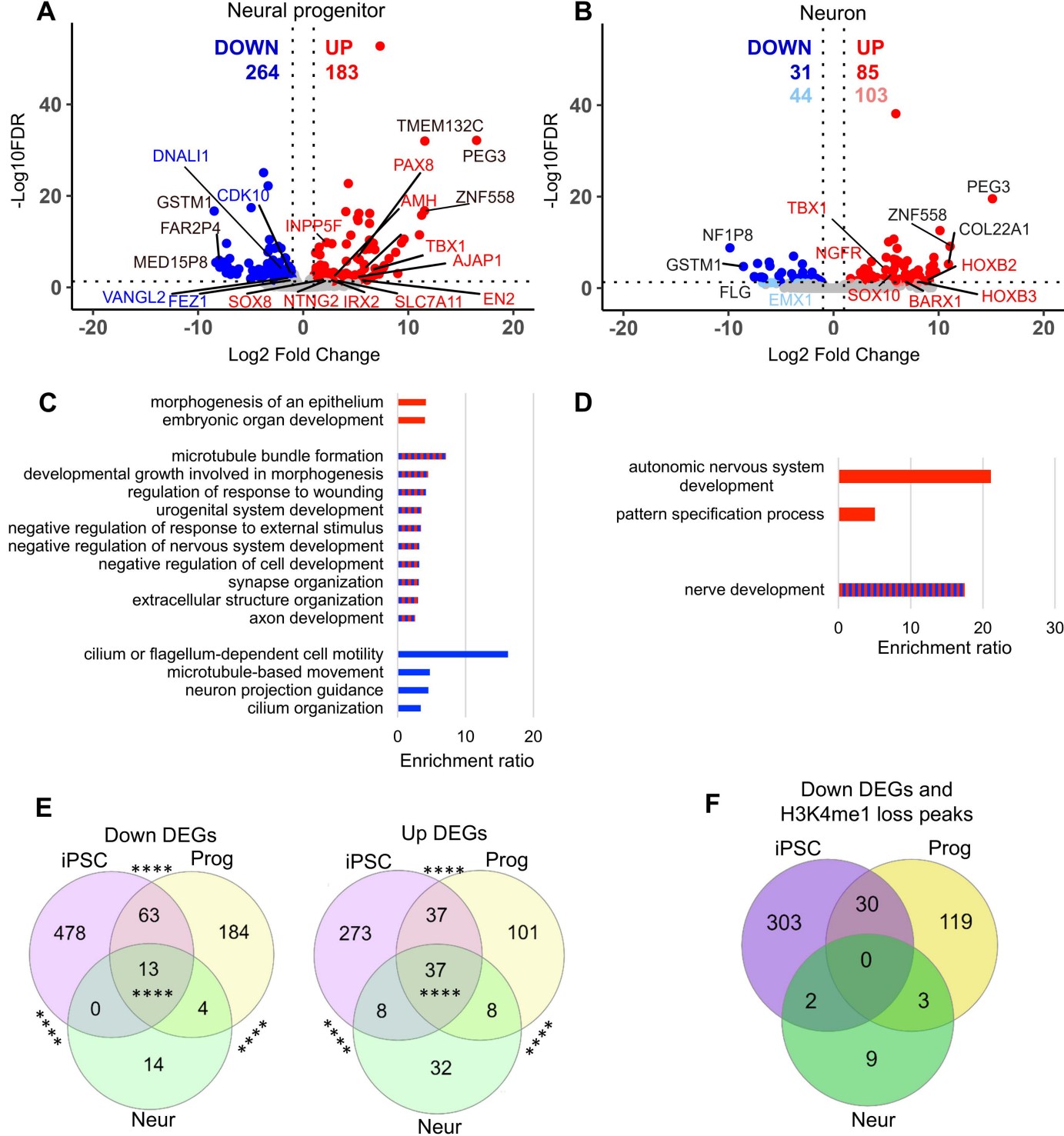

Fig 5. Changes in the transcriptome in KS1 neural progenitors and neurons and comparison of transcriptomic and epigenomic changes during neuronal differentiation. A-B) Volcano plot showing down and up regulated genes on autosomal chromosomes in KS1 (iPSC 3) compared to controls (Ctrl 1) (adjp<0.05, -1>Log2FC>1 in red/blue; adjp<0.1, -1>Log2FC>1 in light-red/light-blue). Top 3 genes with highest or lowest Log2FC are highlighted in black. C-D) WebGestalt was performed on DEGs (adjp<0.1, -1>Log2FC>1) between KS1 and control in neural progenitor and neuronal cells. Gene ontology categories for biological processes are shown in the bar graphs (FDR<0.1) Red bars correspond to upregulated (up) DEGs, blue to

downregulated (down) DEGs and red and blue to both up- and downregulated genes. **E)** Intersection analysis showing DEGs in common between iPSC, Progenitor (Prog) and Neurons (Neur) between KS1 (c.16019G>A) and control (exact test; ****p<0.0001). **F)** Intersection analysis showing Down DEGs and H3K4me1 loss peaks in common between iPSC, Progenitor (Prog) and Neurons (Neur) between KS1 (c.16019G>A) and control.

±100kb of H3K4me1 loss. We identified 30 such genes in common between iPSC and neural progenitor stages, of which 4 (*CNTN4*, *NOVA2*, *GAS8* and *TRIM46*) are involved in axogenesis, neuronal projection guidance and microtubule formation (Fig 5F).

Overall, in our KS1 model we witness a decrease in the number of DEGs with advancing neuronal differentiation.

### The altered KS1 transcriptome is consistent

Most of our neuronal differentiation work was based on a single missense mutant and we observed the biggest differences in the Day 0 (iPSC) and Day 18 (Progenitor) transcriptomes (Fig 5E). We, therefore, validated our results via RNAseq at the three stages (Day 0, Day 18 and Day 30) from our original control and mutant lines, and in two additional controls and patient-derived iPSCs with distinct mutations (c.5527dupA, c.16180G>T). We found several up- and downregulated DEGs in KS1 samples to follow the same gene expression trend (adjusted p-value<0.05; -1<Log2FC>1) at all three stages (Fig 6A, 6C, 6E). Furthermore, 114/909 DEGs (Day 0), 83/447 DEGs (Day 18) and 29/116 DEGs (Day 30) from our original dataset (Figs 3A, 5A, 5B) showed the same gene expression trend in KS1 samples as in the validation Day 0, Day 18 and Day 30 DEGs dataset (S1 Table) (Day 0 Spearman's rho=0.53 and p-value<0.0001; Day 18 Spearman's rho=0.55 and p-value<0.0001; Day 30 Spearman's rho=0.65 and p-value<0.0001) (Fig 6A, 6C, 6E). GO overrepresentation analysis on the DEGs at all three stages in the validation datasets detected biological categories related to KS1 phenotypes (Fig 6B, S5 Table). Additionally, transcription factor enrichment analysis performed on the validation dataset at the three time points also confirmed SUZ12 as a key transcriptional regulator for upregulated DEGs reinforcing its main role in gene regulation in KS1 (S3 Table).

Overall, these results confirm our findings in additional cell lines harbouring distinct *KMT2D* mutations.

## Discussion

Although the underlying disease mechanism of KS1 has been examined previously [7,15,16], several questions about the pathological mechanism of the neurological issues in the disorder remain unanswered. Here, we studied patient-derived iPSC models to interrogate the basis of neurological phenotypes of KS1.

Firstly, haploinsufficiency has been the proposed mechanism for nonsense or frameshift variants in KS1, but it has not been previously proven. We observed ALT:REF read ratios for *KMT2D* to be significantly lower in iPSCs with nonsense or frameshift variants in comparison with the iPSCs with missense variants (Fig 1B). This observation suggests partial nonsense mediated mRNA decay of the mutant transcripts with premature termination codon. Interestingly, the absolute level of REF allele counts was higher in iPSCs with nonsense or frameshift variants than the ones with missense variants, which is likely explained by mRNA decay triggered transcriptional adaptation [21]. The lower levels of reference transcripts in mutant cells with nonsense or frameshift variants in comparison with the controls is also reflected in the protein levels (Fig 1C). The KMT2D protein levels were also significantly lower in iPSCs with the missense variant, even though the overall transcript levels in these lines were not significantly different from controls. This indicates possibly reduced stability of the protein with the amino acid substitution. The p.(Arg5340Gln) variant is located in the FYR-N motif which has been shown to alter the total charge of the domain and reduce KMT2D protein stability [22]. It is possible that other KS1 causing missense variants could produce loss of function through other mechanisms such as altered protein-protein or protein-chromatin interactions or loss of enzyme activity or mislocalization. Although these proposed mechanisms will require validation. As in line with the previous report of decrease in H3K4me3 levels in a KS1 mouse model [7],and with

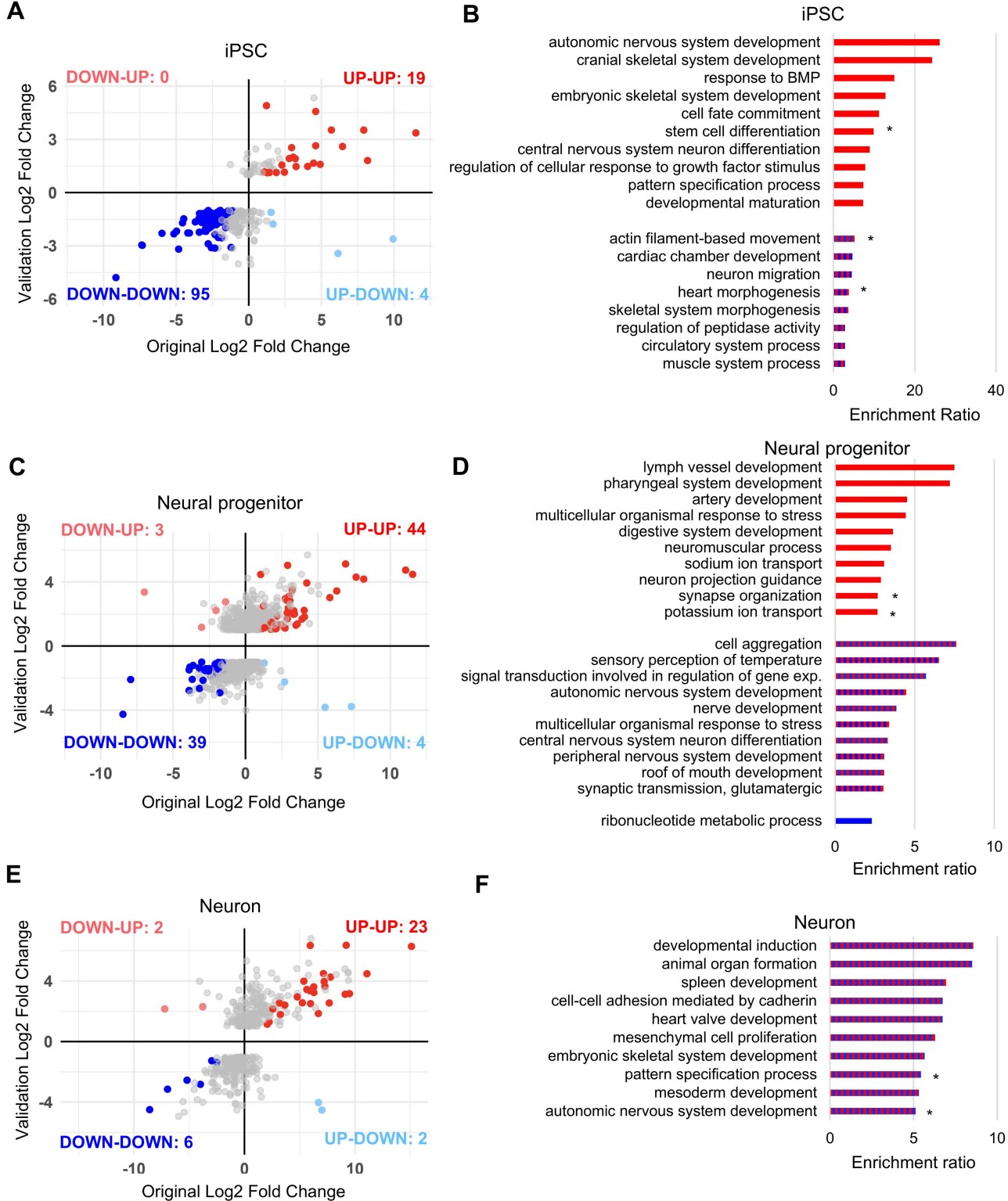

**Fig 6. Validation of the transcriptome in KS1 neural progenitors and neurons.** Comparison of Log2FC of DEGs from the original dataset and validation dataset (KS1 iPSC 1, iPSC 2, iPSC 3 compared to Ctrl 1, Ctrl 2, Ctrl 3) at iPSC (Day 0) (**A**), neural progenitor (Day 18) (**C**) and neuronal stages (Day 30) (**E**). WebGestalt over-representation analysis was performed on DEGs from validation dataset. Gene ontology categories for biological

processes for iPSC (Day 0) (**B**), neural progenitor (Day 18) (**D**) and neurons (Day 30) (**F**) are shown in the bar graphs (FDR<0.05). Red bars correspond to upregulated DEGs, blue to downregulated DEGs and red and blue to both up- and downregulated genes. * indicates GO categories that have been observed in the original dataset.

the known role of KMT2D as a H3K4 mono-methyltransferase [1,3], we found a significant decrease in overall cellular H3K4me1 levels at the iPSC stage (Fig 1D). This is also in agreement with the analysis of 33 KS1 and 36 healthy controls' enhancer signatures related to H3K4me1 and H3K4me2 in peripheral blood mononuclear cells of patients with KS1 [14].

Our experiments also show dysregulation of H3K4me1 and transcriptome landscapes in KS1 iPSCs, neural progenitors and neurons harbouring pathogenic missense variant (p.(Arg5340Gln)) (Figs 2-5). The vast majority of differentially methylated H3K4me1 peaks were losses, which is in line with the known role of KMT2D as a H3K4 methyltransferase [1,2]. Importantly, we find significant correlation between the H3K4me1 and transcriptome dysregulation. This suggests that several of the downregulated DEGs are likely to be direct consequences of loss of H3K4me1 rather than secondary to network disruption. We suspect that these correlations in KS1 may be an underestimate as we only considered regions 100Kb up and downstream of DEGs as well as H3K4me1 peaks with ±1 fold change for our analysis, which ignores potential long-range enhancer activity and potential *trans*-acting effects of H3K4me1 deposition. Furthermore, we observed 'large blocks of dysregulation' rather than it being evenly spread throughout the genome suggesting that certain regions of the genome are more susceptible to loss of KMT2D. Interestingly, with progress in differentiation, we observed an increase in loss of H3K4me1 peaks but decrease in downregulated DEGs. This could reflect higher dependency of H3K4me1 regulation on KMT2D at the pluripotent stage and a less open chromatin state in differentiated cells. Network level adaption could be an alternative explanation. Notably, the controls and KS1 samples in our dataset could not be sex or age matched and, therefore, some of the observed changes could have been influenced by sex differences. The use of non-isogenic controls in our analysis represents a limitation in our study increasing the risk of background-specific effects, which can affect the observed phenotypes. Further studies using isogenic controls will be necessary to confirm the robustness of these results.

The analysis of transcription factor binding sites in our ChIPseq dataset revealed several transcription factors that may be important in KS1 pathology (Figs 2 and 4). KLF [23] and ZNF [24] family transcription factors play crucial roles in cellular processes such as cell growth, differentiation, apoptosis, and response to environmental stimuli. Wilms Tumour 1 (WT1) expression is required in early kidney development and pathogenic mutations in *WT1* gene are associated with genitourinary phenotypes [25] Estrogen-related receptors (ESRR) regulate genes involved in energy homeostasis, including fat and glucose metabolism and mitochondrial biogenesis [26]. Regulatory factor X-5 (RFX5) is involved in regulation of major histocompatibility class II and *RFX5* mutations result in immunodeficiency [27]. IKAROS family zinc finger 1 (IKZF1) is involved in lymphocyte development and pathogenic variants in IKZF1 are associated with immunodeficiency [28]. OTX2 is critical for early embryonic development, particularly in the development of the brain and sensory organs [29]. The effects of retinoic acid on gene expression is mediated by RARG, influencing various biological processes, including development, differentiation, and homeostasis [30]. FOXO6 is involved in regulation of Hippo signalling during craniofacial development, with FoxO6$^{-/-}$ have facial overgrowth [28]. ETS variant (ETV) transcription factors are required for hippocampal dendrite development [31], while TFAP2C is vital for early development, specifically in morphogenesis [32]. Additionally, ETS protein can regulate immune responses and the development of immune-related cells [33].

In our RNAseq datasets, we detected several DEGs important in lineage commitment and neuronal differentiation (Figs 3 and 5). In particular, *PAX8,* important in tissue and organ formation particularly in kidney, thyroid gland and brain [34], is upregulated in KS1 neural progenitor cells. *PAX5,* involved in early neural tube development and regionalization [35], is also upregulated in KS1 iPSC and neural progenitor cells. Its dysregulation in KS1 cells might affect the correct

formation of the neural tube, contributing to the KS1 intellectual disability phenotype. The efficiency of neuronal differentiation in the missense KS1 variants seems to be affected as reflected in a decrease in number of neural progenitors at Day 18 (S9E Fig).

Our data also suggested SUZ12 to be a master regulator of many of the upregulated DEGs in our datasets (S3 Table). SUZ12 is a component of PRC2 required for the H3K27 tri-methylation mark [36]. Further studies will be required to clarify the complex relationships between the chromatin and transcriptional landscape in KS1.

In conclusion, these results establish missense *KMT2D* variant cause KS1 through a loss of function mechanism and result in significant dysregulation of H3K4me1 and the transcriptome landscape across different stages of neuronal differentiation. The H3K4me1 and transcriptome landscapes of the missense *KMT2D* variant causing KS1 are correlated and their dysregulation seem to occur in discreet blocks across the genome. With advancing differentiation from iPSCs to neurons, the dysregulation of the H3K4me1 landscape increase but the number of DEGs decreases. Although the dysregulation of the H3K4me1 landscape is primarily based on studies of a line with missense variant, we expect similar consequences for other types of loss of function variants as observed by the RNAseq analysis (Fig 6).

## Materials and methods

### Ethics statement

Control iPSC (wigw2 - Ctrl 1) and KS1 iPSCs (ierp4 - iPSC 3, aask4 - iPSC 1, oadp4 - iPSC 2) were generated by HipSci (https://www.hipsci.org/#/cohorts/kabuki-syndrome). The generation of hiPSCs has ethical approval from National Institute of Health and Care Research (NIHR) BioResource and members of the HipSci consortium membership committee. National Research Ethics Service (NRES) Committee East of England - Cambridge Central approved the study (REC Ref: 15/EE/0049).

Control iPSCs (JF191b [37] - Ctrl 2, SW171a [38] - Ctrl 3) were generated in house under local ethics (REC 11/H1003/3; IRAS ID 64321 or REC 18/EE0250 IRAS246779).

Formal verbal and written consents were obtained from adults and from the parents of children.

### Generation, validation and culture of induced pluripotent stem cells

Skin biopsies and blood samples were obtained from individuals using standard procedures. Fibroblast cells were cultured in DMEM, 10% foetal bovine serum, 1% L-glutamine at 37°C in a humidified 5% $CO_2$ incubator. Fibroblast cells or peripheral blood mononuclear cells (for Ctrl 2) were reprogrammed into iPSC by using Sendai viral vectors (CytoTune −iPS 2.0 Sendai Reprogramming Kit- ThermoFisher). Control iPSC (wigw2 - Ctrl 1) and KS1 iPSCs (ierp4 - iPSC 3, aask4 - iPSC 1, oadp4 - iPSC 2) were generated by HipSci (https://www.hipsci.org/#/cohorts/kabuki-syndrome) while control iPSCs (JF191b [37] - Ctrl 2, SW171a [38] - Ctrl 3) were generated in house. All the cell lines were generated from different individuals. All cells were grown on Vitronectin N-terminal fragmented (Life Technologies) coated dishes, using Essential 8 medium (StemCELL Technologies). Cells were routinely passaged when 80% confluent using 0.1% EDTA-PBS without $Ca^{2+}$ and Mg. All cell lines were cultured at 37°C in a humidified 5% $CO_2$ incubator.

### Neuronal differentiation

HiPSCs were differentiated to cortical neuron using a protocol adapted from Shi et al 2012 [20]. At the start of the protocol, on Day 0, cells were plated on Matrigel (Corning) and cultured using E8 medium (StemCell Technologies) supplemented with FGF2 (10ng/ml). On Day 1, neuronal differentiation was undertaken using neuronal induction medium supplemented with SMAD signalling inhibitors, Noggin (500ng/ml, Tocris) or Dorsomorphin (Abcam) and SB43154 (10uM, Tocris). On Day 10, cells were passaged and plated on a poly-l-ornithine (Sigma) and murine laminin L2020 (20µg/ml, Sigma) substrate for neuronal maintenance and maturation. On Day 12, small-elongated cells generate rosette structures resembling

early neural tubes which were propagated in culture using FGF2 (20ng/ml) for about 3 days. On Day 18, cells were passaged using Dispase (StemCell Technologies), plated on a poly-l-ornithine and laminin substrate for neuronal maturation and cultured up to Day 30.

## Fluorescence-activated cell sorting (FACS)

Single cell suspensions from neural progenitor (Day 18) and neuronal (Day 30) culture were stained and sorted with an Aria Fusion flow cytometer (BD Biosciences). Cells were blocked in PBS-5%FBS for 30 minutes and incubated with cell surface antibodies, CD44-PE (561858), CD24-FITC (560992) and CD184-APC (560936) for 30 minutes at 4°C. After staining, cells were washed in PBS-5%FBS and filtered through a 0.22 µm filter before sorting and acquisition. All antibodies used for staining were purchased from BD Bioscience. Cells used for RNAseq and ChIPseq were sorted for CD44⁻ CD24⁺ and CD184⁺ for the neural progenitor stage at Day 18 and for CD44⁻ CD24⁺ and CD184⁻ (BD Bioscience) for neurons at Day 30.

## Immunofluorescence staining

Cells grown in 12-well plates were fixed with 4% paraformaldehyde for 15 minutes at room temperature. Samples were permeabilised with 0.5% Triton X-100 in PBS for 10 minutes at room temperature. Then blocking solution (1% BSA in PBS) was applied for 10 minutes at room temperature. Blocking solution was removed and an antibody recognising one of NANOG, OCT4, or SOX2 (4903S, 2890S, 3579S, Life Technologies) was applied overnight at 4°C. After washing three times with PBS, cells were incubated with secondary antibodies of Alexa Fluor 488 goat anti-rabbit (A11008, Invitrogen) for 45 minutes at room temperature in the dark. After washing three times with PBS, samples were stained with DAPI (4083S, Cell signalling technology) for 5 minutes. Finally, samples were washed with PBS. Pictures were taken using an epifluorescence microscope (Olympus U-LH100HG) and cellSens software. Images are representative of three independent experiments.

## Western blot

Nuclear protein samples were isolated from iPSCs using NE-NERT Nuclear and Cytoplasmic Extraction Reagents (ThermoFisher Scientific, Cat#: XJ355181). Then protein samples were quantified using BCA protein assay kit- reducing agent compatible (ThermoFisher Scientific, Cat#: 23250). Electrophoresis was carried out using 10% Bis-Tris Gels and NuPAGE MES SDS Running Buffers (Invitrogen) or 3~8% Tris-Acetate Gels (Invitrogen) and Tris-Acetate SDS Running Buffer (Life Technologies) for KMT2D. Twenty to forty µg of nuclear protein extracts were loaded into the polyacrylamide gel and electrophoresed for 45–60 minutes at 120V. Gels were blotted onto nitrocellulose membranes using Mini iBlot Gel Transfer Stacks Nitrocellulose (Invitrogen) for H3K4me1, me2 and me3, or wet transferred at 200mA over night at 4°C using transfer buffer (pH 8.3) contained glycine, tris base, sodium dodecyl sulfate, methanol and water. After blocking non-specific binding, the membranes were incubated with specific anti-H3K4me1 (ab8895, Abcam), anti-H3K4me2 (C15410035, Diagenode), anti-H3K4me3 (ab8580, Abcam), anti-KMT2D (ab213721, Abcam), and anti-HSP90 (4874S, Cell Signalling) overnight at 4°C. Then, the membrane was incubated with a secondary fluorescent labelled goat anti-rabbit antibody (IRDye 800CW LiCor) and the signal was developed using an Odyssey M imaging machine. Images were analysed using ImageJ by measuring the integrated density of the bands. Three or four experimental replicates were performed.

## Gene expression analysis

Cells were collected by centrifugation and total RNAs were extracted using RNeasy Mini kit (Qiagen) according to the manufacturer's protocol. RNA concentration was measured using a NanoDrop 2000 spectrophotometer (Thermo Scientific). One µg of RNA was reverse transcribed with random hexamers primer (Promega) to generate cDNA using the M-MLV Reverse Transcriptase kit (Promega), according to manufacturer's protocol. Quantitative real-time PCR

(qRT-PCR) reactions were performed in triplicate on a Bio-Rad CFX394 Real Time system (Bio-Rad) using Power SYBR Green PCR Master mix (Applied Biosystems). For each sample, 2 µl cDNA (2 ng/µl) was incubated in a final volume of 10 µl with 5 µl of Power SYBR Green PCR Master mix (Applied Biosystems) and 0.5 µl of both the forward and reverse target-specific primer (10 µM, Sigma). The expression of each target gene was evaluated using a relative quantification approach ($2^{-\Delta CT}$ method) with GAPDH as the internal reference for human genes.

## RNA sequencing and analysis

RNAs were extracted using RNeasy Mini kit (Qiagen) according to the manufacturer's protocol. Library preparation was performed by the Genomic Technologies Core facility at the University of Manchester. Samples were sequenced using NextSeq500 (Dataset 1 – DS1 - UoM, Dataset 3 – DS3 - Validation) or Illumina HiSeq 2000 sequencer (Dataset 2 from HipSci consortium – DS2). Unmapped paired reads were output in BCL format and converted to FASTQ format using bcl-2fastq v2.17.1.14. Prior to the alignment, RNAseq reads were trimmed to remove any adapter or poor-quality reads using Trimmomatic v0.36 with options: "ILLUMINACLIP:Truseq3-PE-2_Nextera-PE.fa:2:30:10 SLIDINGWINDOW:4:20 MIN-LEN:35" [39]. The filtered reads were mapped to the human reference genome (hg19/GRCh37) with comprehensive Gencode v19 genome annotation using STAR v2.5.3a with default options [40]. Duplicate reads were marked using Samtools 1.9 [41] and sorted by name. Reads were attributed to genes using FeatureCounts tool in Subread v2.0.0 [42] in paired-end mode. Prior to the differential gene expression analysis, genes which have less than 10 reads in >3 samples were filtered out, as well as genes on sex chromosomes. Differential gene expression analysis was performed using DESeq2 v1.40.2 using Wald test with Benjamini Hochberg p value adjustment (p adjusted<0.05). EdgeR was used to calculate counts per million. GO over-representation analysis was performed using WebGestalt tool (2019 version). The pluripotency genes in Fig 1E were selected from the gene ontology pluripotency marker category. EnrichR [43,44] was used for transcription regulator enrichment analysis. PEGS [45] tool was used to calculate enrichment of differentially expressed genes adjacent to differential ChIP-Seq H3K4me1 peaks. Statistical analysis of overlap of differentially expressed genes across timepoints was performed using SuperExactTest package within R v4.3.1. To assess clustering of DEGs in a statistical manner, we performed hypergeometric test for over-enrichment of DEGs in 10Mb genomic bins across autosomes (excluding patches, haplotypes and undefined contigs).

For validation heatmap generation and gene-based Z-score transformation of RNASeq count data pheatmap package within R v4.3.1 was used. Separate DGE analysis was performed for validation dataset 3 (-1<Log2 FC>1, adjusted p-value <0.05). Then DEGs from Dataset 1 were intersected with DEGs from validation Dataset 2 or 3 and plot them in a heatmap representation.

## Chromatin immunoprecipitation sequencing and analysis

ChIPseq has been performed using the True MicroChIP kit from Diagenode following the manufacturer's protocol. Briefly, cells were collected and DNA-protein was cross-linked using 36.5% formaldehyde. Then cells were lysed and chromatin was sheared by sonication (Bioruptor NextGen). Sheared chromatin was incubated with H3K4me1 Antibody (5326, Cell signalling) overnight at 4°C. Library preparation was performed by the Genomic Technologies Core facility at the University of Manchester.

Unmapped paired-end sequences from an Illumina HiSeq4000 sequencer were output in BCL format and converted to FASTQ format using bcl2fastq v2.20.0.422, during which adapter sequences were removed. Unmapped paired-reads of 76 bp were interrogated using a quality control pipeline consisting of FastQC v0.11.3 (http://www.bioinformatics.babraham.ac.uk/projects/fastqc/) and FastQ Screen v0.9.2 [46]. Prior to the alignment, reads were trimmed to remove any adapter or poor-quality sequence using Trimmomatic v0.36 [39], and truncated at a sliding 4 bp window, starting 5', with a mean quality <Q20, then removed if the final length was less than 35 bp. The filtered paired-reads were mapped to the human reference sequence analysis set (hg19/GRCh37) from the UCSC browser, using Bowtie2 v2.3.0 [47]. Mapped reads were

then filtered using Samtools v0.1.19 to retain only high-confidence concordant pairs (-f 2 -q30), followed by the removal of reads mapping to the mitochondrial genome and to unassembled parts of the reference genome. Peak calling was performed using MACS2 v2.1.2 [48] using the parameters ' --format BAMPE --gsize hs --keep-dup 1 --bdg --SPMR --qvalue 0.05'. Candidate regions were further filtered by fold enrichment score. Differential binding analysis was run on peaks which were found in at least 2 samples using DiffBind v2.10.0 on R v4.3.1 (http://bioconductor.org/packages/release/bioc/vignettes/DiffBind/inst/doc/DiffBind.pdf [49] and R Core Team 2023). The input was the binding region coordinates and associated q-values from MACS2, and mapped reads for each sample. Differential binding analysis was performed based on the comparison of KS1 versus control samples with Benjamini-Hochberg adjusted p-value threshold of 0.05. Bedtools [50] intersect tools was used to intersect enhancer regions from EpiMap repository [51] based on iPSC DF19.11, BSS01370 Neural Progenitor or BSS01366 Neural cells with H3K4me1 regions from ChIP-Seq analysis with minimum intersect ratio of 0.5 in relation to enhancer region. Deeptools v3.4.3 [52] was used to compare the distribution and intensity of H3K4me1 peaks across samples in different genomic regions. BamCompare was used to subtract ChIP input reads from each sample, whilst being simultaneously normalised to bins per million (BPM) with bin size of 20. ComputeMatrix created a matrix of reads for the normalised reads of each sample relative to the sites of interest used for visualisation, which was then passed to plotHeatmap for visualisation of read distribution within regions of interest. HOMER software was used for annotation of peaks [53].

Analysis of ChIP-seq data between different cell time points was performed using ChIPpeakAnno v3.36.0 package within R v4.3.1.

For motif analysis, unique peak summits from samples within differential H3K4me1 regions were extracted using Bedtools [50] intersect. Flanking regions of ±250 bp were added to the peak summits (Bedtools [50] slop) and fasta sequences within those regions were extracted using Bedtools [50] getfasta. MEME-TomTom within MEME-ChIP was used to search JASPAR CORE [54] vertebrates, UniProbe Mouse [55] and Jolma 2013 Human and Mouse databases [56] for predicted motifs within differential H3K4me1 regions.

## Statistical analysis

Two-tailed T-test was performed to analyse the differences in mean expression values of KMT2D and intensity of H3K4me1/2/3. Hypergeometric test was performed to assess over-representation of DEGs per chromosome and per chromosomal bins. Two-tailed Fisher's test was performed to analyse the association of H3K4me1 loss within the enhancer regions. Spearman correlation test was performed to compare correlation of Log2 Fold Changes of DEGs from original dataset (DS1 - UoM) and Log 2 Fold Changes calculated from validation dataset (DS3 - UoM).

## Supporting information

**S1 Table. List of samples used in the performed experiments.** Samples name, number, sex and mutation are listed.
(XLSX)

**S2 Table. List of HipSci cell lines used in the study.**
(XLSX)

**S3 Table. Consensus Transcription factor (TF) analysis.** CHEA and ENCODE consensus transcription factor (TF) analysis for upregulated (UP), downregulated (DOWN) and both up and downregulated (UP and DOWN) DEGs (adjp<0.05) at the three different differentiation stages and in the validation dataset.
(XLSX)

**S4 Table. Hypergeometric analysis of gene-peak associations across different genomic distances.** Association of DEGs and differential H3K4me1 peaks in iPSCs, Neural Progenitors and Neurons. Associations with p-value<0.05 of

downregulated DEGs and loss of H3K4me1 are in blue, and associations with p-value < 0.05 of upregulated DEGs and gain of H3K4me1 are in red.
(XLSX)

**S5 Table. WebGestalt over-representation analysis on validation dataset.** Gene ontology categories for biological processes are listed. Categories highlighted in yellow have been observed in the original dataset.
(XLSX)

**S1 Fig. Integration of RNAseq datasets.** PCA plot showing distinct clustering of the two groups (control: blue; KS1: red).
(TIF)

**S2 Fig. Analysis of *KMT2D* expression A)** qRT-PCR analysis of *KMT2D* transcript level relative to *GAPDH* in control and KS1 iPSCs (frameshift (c.5527dupA, p. (Pro1849Ter)), missense (c.16019G>A, p. (Arg5340Gln)), and nonsense (c.16180G>T, p. (Glu5394Ter)) variants (n = 3)). **B)** Mutant read visualization using IGV. **C)** Bar graph showing percentage of REF (reads with reference – no mutation; blue for control and light blue for KS1) and ALT (reads with mutation, orange) (DS2 iPSC1 c.5527dupA, DS2 iPSC 2.1 c.16180G>T, DS2 iPSC 2.2 c.16180G>T, DS2 iPSC 3 c.16019G>A, DS1 iPSC 3.1 c.16019G>A, DS1 iPSC 3.2 c.16019G>A, DS1 iPSC 3.3 c.16019G>A).
(TIF)

**S3 Fig. Analysis of KMT2D and methylation marks.** Full images of Western blots for KMT2D (**A**), H3K4me1 (**B**), H3K4me2 (**C**), H3K4me3 (**D**).
(TIF)

**S4 Fig. Analysis of H3K4 methylation using DepMap.** Volcano plot showing the level of histone marks in cell lines carrying *KMT2D* mutations using DepMap portal (https://depmap.org/portal). H3K4me1 and H3K4me2 levels are significantly reduced.
(TIF)

**S5 Fig. Validation analysis for pluripotency in KS1 and control iPSC lines. A)** Immunofluorescence staining for OCT4, NANOG and SOX2 in iPSCs. **B)** qRT-PCR analysis of *POU5F1*, *NANOG* and *SOX2* transcript level relative to *GAPDH* in control (Ctrl 1, Ctrl 2, Ctrl 3) and KS1 iPSCs (iPSC 1, iPSC 3, iPSC 2) (n = 3).
(TIF)

**S6 Fig. Analysis of H3K4me1 level in different genomic regions in iPSCs. A)** Heatmaps of H3K4me1 regions mapped to total (n = 81,930 in black; left), H3K4me1 loss (n = 3647 in blue; right) and gain (n = 349 in red; right) Epi-Map enhancer regions based on all cell-types. Top TF motifs predicted in differential H3K4me1 peaks (loss and gain of H3K4me1 in binding regions) between KS1 and control in **B)** cell–type independent enhancers and **C)** cell-type independent promoters. Number (n) represents the number of times the motif was found within the unique sequences underlying the differential H3K4me1 regions.
(TIF)

**S7 Fig. Association analysis of loss of H3K4me1 in KS1 iPSC, progenitors and neurons.** Analysis of association of H3K4me1 loss with enhancer regions at day 0 iPSC stage (**A**), at day 18 neural progenitor stage (**B**) and at day 30 neuronal stage (**C**) with percentage of loss per region (two-tailed fisher's exact test). Not loss represents peaks that were found not to be statistically significantly downregulated.
(TIF)

**S8 Fig. Analysis of the transcriptome in KS1 iPSCs. A)** PCA plot showing distinct clustering of the two groups (control: blue; KS1 (c.16019G>A): red) at the iPSC stage in DS1-UoM. **B)** PCA plot showing all the three cell stage Blue

circle show iPSC, green progenitors and orange neuronal cells (control: blue; KS1: red). **C)** Bar graph showing percentage of DEGs per total gene on chromosome in KS1 compared to controls. Upregulated genes are in red and downregulated genes are in blue. Hypergeometric test was performed to assess the enrichment of DEGs in each chromosome (*FDR<0.05; ****FDR<0.0001; hypergeometric test). **D)** Volcano plot showing DEGs within ±100kb of H3K4me1 loss regions.
(TIF)

**S9 Fig. Analysis of H3K4me1 level in KS1 neural progenitor cells. A**) Representative phase contrast images of control and KS1 neural progenitor (Day 18) morphology. Rosettes are highlighted by yellow arrowheads. **B**) Heatmap of H3K4me1 peaks (black), H3K4me1 loss regions (blue), and H3K4me1 gain regions (red) for neural progenitors. **C)** Volcano plot showing loss (blue) and gain (red) in H3K4me1. **D)** Top TF motifs predicted in differential H3K4me1 peaks (loss and gain of H3K4me1 in binding regions) between KS1 and control in cell–type independent enhancers. Number (n) represents the number of times the motif was found within the unique sequences underlying the differential H3K4me1 regions. **E)** Flow cytometry analysis for neural progenitor cells (CD44$^-$ CD184$^+$ CD24$^+$) at Day 18 showing representative flow plots and scatter plot for quantification.
(TIF)

**S10 Fig. Analysis of H3K4me1 level in different genomic regions in neuronal cells. A)** Representative phase contrast images of control and KS1 neuronal morphology **B)** Heatmap of H3K4me1 peaks (black), H3K4me1 loss regions (blue), and H3K4me1 gain regions (red) for neurons. **C)** Volcano plot showing loss (blue) and gain (red) in H3K4me1. Top TF motifs predicted in differential H3K4me1 peaks (loss and gain of H3K4me1 in binding regions) between KS1 and control in **D)** cell–type independent enhancers and **E)** cell-type independent promoters. Number (n) represents the number of times the motif was found within the unique sequences underlying the differential H3K4me1 regions.
(TIF)

**S11 Fig. Analysis of the transcriptome in KS1 neural progenitor cells. A)** Bar graph showing percentage of DEGs per total gene on chromosome in KS1 compared to controls. Upregulated genes are in red and downregulated genes are in blue. Hypergeometric test was performed to assess the enrichment of DEGs in each chromosome (*FDR<0.05; ****FDR<0.0001; hypergeometric test). **B)** Volcano plot showing down and up DEGs within ±100kb of differentially lost H3K4me1 regions. **C)** Scatter plot showing change in expression and change in H3K4me1 within ±100kb of DEGs.
(TIF)

**S12 Fig. Analysis of the transcriptome in KS1 neuronal cells. A)** Bar graph showing percentage of DEGs per total gene on chromosome in KS1 compared to controls. Upregulated genes are in red and downregulated genes are in blue. Hypergeometric test was performed to assess the enrichment of DEGs in each chromosome (*FDR<0.05; **FDR<0.01; hypergeometric test). **B)** Volcano plot showing down and up DEGs within ±100kb of differentially lost H3K4me1 regions. **C)** Scatter plot showing change in expression and change in H3K4me1 within ±100kb of DEGs.
(TIF)

**S13 Fig. Positional association of H3K4me1 loss peaks and DEGs. A-B)** Karyoplot showing chromosomal distribution of ±100kb H3K4me1 loss peaks (green bars) and downregulated (blue circles) or upregulated (red circles) DEGs in **(A)** progenitor or **(B)** neuronal cells. Horizontal bars represent regions of statistically significant clustering of downregulated (blue bar) or upregulated (red bar) DEGs (Hypergeometric test; FDR<0.05).
(TIF)

**S1 Appendix. List of DEGs from iPSC analysis.**
(XLSX)

**S2 Appendix. List of DEGs from neural progenitor cell analysis.**
(XLSX)

**S3 Appendix. List of DEGs from neuronal cell analysis.**
(XLSX)

**S4 Appendix. List of differential H3K4me1 peaks from iPSC analysis.**
(XLSX)

**S5 Appendix. List of differential H3K4me1 peaks from neural progenitor cell analysis.**
(XLSX)

**S6 Appendix. List of differential H3K4me1 peaks from neuronal cell analysis.**
(XLSX)

## Acknowlegments

We thank KS1 individuals and their families for providing the samples. We also thank Julieta O'Flaherty and Steven Woods for the control iPSCs (JF191b [37], SW171a [38]) and the Genomic Technologies Core Facility at the University of Manchester for RNA and ChIP library preparation and sequencing. This study makes use of control and KS1 iPSCs and RNAseq data generated by the HipSci consortium (www.hipsci.org), funded by The Wellcome Trust and the MRC.

## Author contributions

**Conceptualization:** Sara Cuvertino, Susan J. Kimber, Siddharth Banka.

**Data curation:** Sara Cuvertino, Evgenii Martirosian.

**Formal analysis:** Ian J. Donaldson.

**Funding acquisition:** Sara Cuvertino, Adam Stevens, Andrew D. Sharrocks, Susan J. Kimber, Siddharth Banka.

**Investigation:** Sara Cuvertino, Evgenii Martirosian, Kedar Bhosale, Peiwen Cheng.

**Project administration:** Sara Cuvertino, Evgenii Martirosian, Susan J. Kimber, Siddharth Banka.

**Resources:** Adam Stevens, Susan J. Kimber, Siddharth Banka.

**Validation:** Kedar Bhosale.

**Visualization:** Sara Cuvertino, Evgenii Martirosian, Peiwen Cheng.

**Writing – original draft:** Sara Cuvertino, Evgenii Martirosian.

**Writing – review & editing:** Sara Cuvertino, Evgenii Martirosian, Terence Garner, Adam Jackson, Adam Stevens, Andrew D. Sharrocks, Susan J. Kimber, Siddharth Banka.

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
