## [Decision Letter · Decision Letter 0]

30 Mar 2025

PGENETICS-D-25-00138

Epigenome and transcriptome changes in KMT2D-related Kabuki syndrome Type 1 iPSCs, neuronal progenitors and cortical neurons

PLOS Genetics

Dear Dr. Martirosian,

Thank you for submitting your manuscript to PLOS Genetics. After careful consideration, we feel that it has merit but does not fully meet PLOS Genetics's publication criteria as it currently stands. Therefore, we invite you to submit a revised version of the manuscript that addresses the points raised during the review process. 

Please submit your revised manuscript within 60 days May 29 2025 11:59PM. If you will need more time than this to complete your revisions, please reply to this message or contact the journal office at plosgenetics@plos.org. Please include the following items when submitting your revised manuscript:

We look forward to receiving your revised manuscript.

Kind regards,

Marnie E. Blewitt

Section Editor

PLOS Genetics

Gregory Cooper

Section Editor

PLOS Genetics

Aimée Dudley

Editor-in-Chief

PLOS Genetics

Anne Goriely

Editor-in-Chief

PLOS Genetics

**Journal Requirements:**

At this stage, the following Authors/Authors require contributions: Sara Cuvertino, Evgenii Martirosian, Peiwen Cheng, Terence Garner, Ian J. Donaldson, Adam Jackson, Adam Stevens, Andrew D. Sharrocks, Susan J. Kimber, and Siddharth Banka. Please ensure that the full contributions of each author are acknowledged in the "Add/Edit/Remove Authors" section of our submission form.

The list of CRediT author contributions may be found here: https://journals.plos.org/plosgenetics/s/authorship#loc-author-contributions

https://journals.plos.org/plosgenetics/s/submission-guidelines#loc-parts-of-a-submission

4) We do not publish any copyright or trademark symbols that usually accompany proprietary names, eg ©,  ®, or TM  (e.g. next to drug or reagent names). Therefore please remove all instances of trademark/copyright symbols throughout the text, including:

- TM on page: 12.

5) Thank you for including an Ethics Statement for your study. Please include:

i) A statement that formal consent was obtained (must state whether verbal/written) OR the reason consent was not obtained (e.g. anonymity). NOTE: If child participants, the statement must declare that formal consent was obtained from the parent/guardian.].

6) Please ensure that all Figure files have corresponding citations and legends within the manuscript. Currently, Figures 6, and 7 in your submission file inventory do not have in-text citations. If the figures are no longer to be included as part of the submission, please remove them from the file inventory.

7) We have noticed that you have cited Tables  1, and 2 in the manuscript file but there are no corresponding tables in the manuscript.  Please amend your manuscript to include these tables noting that tables should not be uploaded as individual files.

8) We have noticed that you have uploaded Supporting Information files, but you have not included a list of legends. Please add a full list of legends for your Supporting Information files after the references list.

9) Thank you for stating "The data that support the findings of this study are publicly available from NCBI GEO with the identifier(s) GSE289158 and GSE289159." Please note that, though access restrictions are acceptable now, your entire minimal dataset will need to be made freely accessible if your manuscript is accepted for publication. This policy applies to all data except where public deposition would breach compliance with the protocol approved by your research ethics board. If you are unable to adhere to our open data policy, please kindly revise your statement to explain your reasoning and we will seek the editor's input on an exemption.

10) Please amend your detailed Financial Disclosure statement. This is published with the article. It must therefore be completed in full sentences and contain the exact wording you wish to be published.

3) If any authors received a salary from any of your funders, please state which authors and which funders.

**Reviewers' comments:**

Reviewer's Responses to Questions

Reviewer #1: Cuvertino et al., perform a comprehensive mechanistic assessment of changes in gene expression and histone methylation across iPSC, neuronal progenitor and neuronal cell types. They address how changes in histone methylation may underlie the traits associated with the neurodevelopment Kabuki Syndrome.

Overall, this study is very interesting and bridges an important gap in the knowledge as to the underlying molecular mechanisms of Kabuki syndrome. I suggest the following changes to the test to improve this paper:

(i) The authors state that the number of differential H3K4me1 peaks between KS1 and control samples is reduced through differentiation and conclude that this reflects a difference in the requirement of KS1 during the differentiation process. An alternative interpretation is that chromatin compaction increases with differentiation, as cells become more specialised and only express genes required for their lineage, as opposed to the more open and permissive chromatin state in iPSCs and stem cells. As such, the number of regions able to be methylated on H3K4 will be reduced as cells differentiate, and this may underly the reduced number of H3K4me1 peaks in the progenitors and neurons in this study, relative to the iPSCs.

(ii) Do the KS1 and control iPSCs differentiate into neurons with different efficiencies? The images in Supp Figure 8 show that they have markedly different morphologies at day 18. A comment on the efficiency of neuronal differentiation across KS1 lines relative to controls should be added to the results or discussion.

(iii) Supplemental Table 1 shows that the iPSC control lines are from 2 male donors and 1 female, while the KS1 samples are from 2 females and 1 male. The potential for sex-associated differences across datasets should be discussed. To this point, iPSC line 1 and 3 show much higher levels of Oct4 and Nanog expression by RTqPCR, compared to iPSC line 2. iPSC lines 1 and 3 are female, while iPSC line 2 is male, such that sex-specific differences may underlie this result. Although, this could also be explained by iPSC line 2 carrying a nonsense mutation rather than a missense or frameshift as in the other two lines. Regardless, a discussion on potential sex differences should be added to the discussion and I suggest that the authors reword their comment about pluripotency gene expression not being different in these cells as it is markedly increased at Oct4 and Nanog in 2 of their 3 cell lines.

(iv) I would suggest replacing Figure 1 B with Supplemental Figure 2A. Supplemental Figure 2A is a bit easier to interpret.

(v) Which control iPSC line(s) were used in ChIP-seq experiments? The authors do not specify and whether this was a male or female line would be relevant to add to the discussion.

(vi) Were the regions with gained H3K4me1 in the ChIP-seq datasets notable in any way? A comment should be added to the results section.

(vii) Supplemental files with the differentially expressed genes and regions of differential H3K4me1 abundance across datasets should be provided. Also, it is not clear if these datasets will be uploaded to public platforms.

Reviewer #2: This manuscript explores the transcriptomic and epigenomic mechanisms underlying Kabuki syndrome 1 (KS1) in three KS1 patient-derived induced pluripotent stem cells (iPSC) lines. This would be a tremendous resource for the neurodevelopmental disorder field and further our understanding into disease mechanisms of KS1. The authors have addressed questions relevant to KS1 and conducted appropriate experiments to validate their iPSC models in relation to current knowledge of KMT2D function. It’s evident that the overarching goal of this study was to compare and contrast the transcriptome and epigenome of different KS1 mutations across pluripotent stem cells and more disorder-relevant cell types including neural progenitors and mature neurons. In it’s current state, this manuscript falls short of these aims and lacks sufficient detail and clarity regarding the datasets used between experiments and rationale for focusing primarily on the missense variants in neural cell types (detailed below). This reviewer recommends the following major corrections:

Clarity around datasets, controls, replicates and lines

• Why were different datasets used for experiments and why do some iPSC line appear across multiple datasets but others do not.

• Additional details regarding the control lines used is required (are they isogenic corrected or from healthy individuals).

• There are 6 controls listed in supp. Table 1 across three datasets, are these all from different individuals or are some of these matching? This needs to be stated.

• What denotes an iPSC line vs a replicate? It appears multiple lines/replicates have been generated from a single individual. In it’s current state, this is confusing and lacks sufficient detail to interpret the results sufficiently. Are lines generated via an independent induction experiment or are they different clones from the same experiment? Are replicates internal replicates or experimental replicates?

• If the author is merging independent biological replicates and experimental replicates for analysis, how are they accounting for these variables statistically?

Discordance in experiments across the study

The authors initially explored the impact of three Kabuki Syndrome 1, KMT2D patient variants – a missense, a nonsense and frameshift variant – on H3K4 methylation and KMT2D expression in pluripotent stem cells. The remainder of the experiments in this study were less powered and only focused on the missense mutation. It is unclear why the authors did not pursue a comparison of different KS1 mutations across the experiments in this manuscript and why the missense mutation was chosen for further analysis, especially in light of the differing mRNA results in Fig 1B which may allude to a differing mechanism and impact on transcription and H3K4 methylation in comparison to other mutations.

The authors partially address this in the last section by conducting an additional validation RNAseq experiment with two control lines and the frameshift variant in neural progenitors. DEG comparison with the missense variant experiment had a moderate positive correlation, suggestive of a similar trend in gene expression changes between the missense and frameshift mutations. The nonsense mutation is however absent from this analysis and is only included within the first RNAseq experiment in iPSC’s.

The authors should consider changing the focus of this study to primarily reflect their experiments which are concentrated on assessing transcriptomic and epigenetic differences in the missense mutation in different cell types in comparison to healthy control lines to improve the clarity and comprehension of the manuscript.

Minor corrections:

Figure 1

• Western blot quantification graphs in panel C and D need units for the Y-axis.

• In panel B, the graphical representation comparing ALT KMT2D transcripts between missense, nonsense and frameshift mutations and the associated significant bar is confusing and separating these into their genotypes would be helpful to better interpret the results.

• In the figure legend, you need to state the statistical analysis conducted and the n for each group.

• Panel E: It’s unclear how these pluripotency markers were chosen. A geneset enrichment analysis with a pluripotency geneset would be more appropriate.

Supp. Table 1.

• Having a combined column for # Samples/Sample name for Controls is confusing and should be separated. It is unclear whether controls from dataset 1 and 2 come from Ctrl 1, 2 or 3 individuals or different individuals. This would be more clear if the columns were separated and this information provided

• Number of samples should be in the sex column of DS2 controls.

• Line 186: SUZ12 was identified only in DEGs from the RNAseq dataset which were upregulated, but not in downregulated peaks. Additionally you did not identify SUZ12 in your motif analysis at H3K4me1 peaks. This should be more accurately reflected in your summary of the data from these experiments.

Figure 3.

• It is unclear why the authors have chosen to perform a DEG analysis on just the missense variant iPSC lines and not the other variants, especially considering they have these datasets based on the Fig 1. results. It would be interesting to conduct a DEG analysis across mutants in comparison to the control lines to determine if common or different genes and genesets are dysregulated.

• Line 191: A better rationale for assessing neural progenitors is needed beyond stating KS1 clinical phenotypes.

• Line 224: these numbers should accurately reflect the numbers in Fig. 4I.

Figure 4.

• In the figure legend an independent description of what cell types panel E and F represent should be provided.

• The total loss peaks and gain peaks for each cell type in panel I do not match the numbers in panel E and F. For example, in panel I the total loss peaks for neurons is 21,584 (20321+402+74+787), but in panel F the total loss peaks is 21,586.

• Abbreviations such as Neur and Prog should be defined in the figure legend.

Figure 5.

• To be more comparable with the iPSC datasets, similar representation of analyses should be displayed. Therefore A karyoplot like in Fig 3C should be included for neural progenitors and neurons.

• Abbreviations such as Neur and Prog should be defined in the figure legend.

• Line 266: ‘into’ should be ‘in’

• Supp Table 1 refers to 3 datasets – DS1, DS2 and DS3, yet in the text you also refer to a validated dataset (line 280 and 604 for example) which is confusing. This need to be clarified and added to the Supp Table 1.

• Line 313: The authors need to clarify that this was only demonstrated for the missense mutation.

• Line 356: This is limited to the missense mutation based on the results from this study. Discussion around how the results of the missense mutation across cell types may inform on the mechanisms of the frameshift and nonsense variants in neural tissue and genomic H3K4me1 would be more accurate.

Reviewer #3: This is an interesting study evaluating transcriptomic and epigenetic changes in iPSC-derived cell models from Kabuki syndrome type 1 (KS1) patients. The authors track the epigenome and transcriptome across three neuronal differentiation stages, identifying significant disruptions in KMT2D expression, H3K4 methylation, and enhancer-associated histone marks. They report that epigenomic alterations accumulate during differentiation, while the number of differentially expressed genes decreases. The study highlights the role of Polycomb-mediated repression and transcription factor binding changes in KS1 pathogenesis.

However, some issues regarding the generalizability of the data across different iPSC lines need to be addressed to enhance the study’s impact and clarity before publication.

Specific Comments

1. In this study, the authors performed transcriptomic and epigenetic profiling of iPSC-derived cell models (iPSCs, neuronal progenitors, and neurons) to investigate molecular changes in KS1. They utilized three different iPSC lines from KS1 patients. However, the core analyses were conducted using only one KS1 iPSC line with a missense mutation (c.16019G>A, p.(Arg5340Gln)), making it difficult to generalize the proposed mechanism (Polycomb-mediated repression and transcription factor binding changes). This issue is compounded by the lack of quality control data for the iPSC lines.

a. Figures 2–5: It is unclear what control was used in the comparison with the c.16019G>A (p.(Arg5340Gln)) line—was it a single control line or all controls?

b. iPSC Quality Control: There is no data on the quality control (e.g., karyotyping) of the KS1 iPSCs, which is essential for ensuring the reliability of the findings.

c. Figure 5G: The overlap of differentially expressed genes (DEGs) between the three KS1 iPSC lines is only briefly explored, and only between iPSCs and neuronal progenitors. The observed overlap between all three lines is of high interest, as these likely represent the most robust KMT2D-dependent changes. However, the presented data are sparse and feel incomplete, with no corresponding methylation data.

2. Although several mechanisms are being proposed and discussed there no single validation presented.

3. Figure 1D: The authors should use H3 as a loading control to provide a more accurate estimation of H3 methylation changes. The number of technical and biological replicates for the western blot is unclear—does "N=3" refer to three independent iPSC lines, or does it mean the western blot was performed once for each line?

a. HSP90 Blot (H3K4me3): The blot appears problematic, as the protein seems to have been retained in the wells instead of migrating through the gel.

b. There is considerable variation in the western blots that is not reflected in the quantification. How do the authors explain this? They should provide full western blot images for clarity.

Minor Comments

Figure 1B: The description in the text is unclear. Please rephrase for better comprehension.

Figure 1C: Why would a missense mutation lead to decreased KMT2D levels? Do the authors have any explanation?

Figure Presentation: The manuscript, including figures, appears somewhat sloppy. Figures are not always clear, and their presentation could be improved.

Figures 5 & 6: These figures are not referenced in the text and should likely be converted into tables for clarity.

**Have all data underlying the figures and results presented in the manuscript been provided?**

Reviewer #1: **No: ** Results of RNA-seq and ChIP-seq experiments have not been provided as supplemental files

Reviewer #2: **No: ** RNAseq and ChIPseq datasets presented in this study do not have a public repository accession number provided. It is highly recommended that these datasets are made publicly available prior to publication. Supplying uncropped western blot images as supplementary figures is also highly recommended.

Reviewer #3: Yes

PLOS authors have the option to publish the peer review history of their article (what does this mean? ). If published, this will include your full peer review and any attached files.

**Do you want your identity to be public for this peer review?** For information about this choice, including consent withdrawal, please see our Privacy Policy .

Reviewer #1: No

Reviewer #2: **Yes: ** Jordan Wright

Reviewer #3: No

**Figure resubmission:**
---

## [Decision Letter · Decision Letter 1]

30 Jun 2025

PGENETICS-D-25-00138R1

Epigenome and transcriptome changes in KMT2D-related Kabuki syndrome Type 1 iPSCs, neuronal progenitors and cortical neurons

PLOS Genetics

Dear Dr. Martirosian,

Thank you for submitting your manuscript to PLOS Genetics. After careful consideration, we feel that it has merit but does not fully meet PLOS Genetics's publication criteria as it currently stands. Therefore, we invite you to submit a revised version of the manuscript that addresses the points raised during the review process. While we do not believe additional bench work is required, that changes to the text and the additional RNA-seq analyses described by the reviewer of the current version need to be addressed.

Please submit your revised manuscript within 30 days Jul 30 2025 11:59PM. If you will need more time than this to complete your revisions, please reply to this message or contact the journal office at plosgenetics@plos.org. Please include the following items when submitting your revised manuscript:

We look forward to receiving your revised manuscript.

Kind regards,

Gregory Cooper

Section Editor

PLOS Genetics

Aimée Dudley

Editor-in-Chief

PLOS Genetics

Anne Goriely

Editor-in-Chief

PLOS Genetics

**Journal Requirements:**

Please amend your detailed Financial Disclosure statement. This is published with the article. It must therefore be completed in full sentences and contain the exact wording you wish to be published. Please ensure that the funders and grant numbers match between the Financial Disclosure field and the Funding Information tab in your submission form. Note that the funders must be provided in the same order in both places as well.

State what role the funders took in the study. If the funders had no role in your study, please state: "The funders had no role in study design, data collection and analysis, decision to publish, or preparation of the manuscript.".

**Reviewers' comments:**

Reviewer's Responses to Questions

**Comments to the Authors:**

Reviewer #1: All of my comments have been thoroughly addressed. The authors have done a clear and thorough job of addressing all points.

Reviewer #3: Unfortunately, several of the most critical comments remain inadequately addressed in the rebuttal, leaving key concerns unresolved and diminishing confidence in the rigor and interpretability of the study. While the authors provided responses, many of them either sidestepped the core issues or deferred them to future work without proper justification.

1a. In this study, the authors performed transcriptomic and epigenetic profiling of iPSC-derived

cell models (iPSCs, neuronal progenitors, and neurons) to investigate molecular changes in

KS1. They utilized three different iPSC lines from KS1 patients. However, the core analyses

were conducted using only one KS1 iPSC line with a missense mutation (c.16019G>A,

p.(Arg5340Gln)), making it difficult to generalize the proposed mechanism (Polycombmediated

repression and transcription factor binding changes).

The decision to use only one non-isogenic control remains highly problematic and random, especially in a context where genetic background is a known confounder. The authors have not added any substantial clarification or justification for this design choice. The authors should at least incorporate a dedicated paragraph in the Discussion to highlight the significant limitations imposed by their case-control design. As it stands, the study lacks the statistical power and experimental design required to support any strong or definitive conclusions.

1b. iPSC characterisation informations

Details regarding the control setup should be clearly specified in the Materials and Methods section.

1c. Figure 5G: The overlap of differentially expressed genes (DEGs) between the three KS1

iPSC lines is only briefly explored, and only between iPSCs and neuronal progenitors. The

observed overlap between all three lines is of high interest, as these likely represent the most

robust KMT2D-dependent changes. However, the presented data are sparse and feel

incomplete, with no corresponding methylation data.

The authors' response that further analysis will be part of future work is not acceptable in this context. They already have the data available, and given the limitations of their case-control setup, it is crucial to robustly identify differentially expressed genes (DEGs) within the current study. This is not a matter for future investigation but a requirement for the current manuscript to meet basic standards of scientific rigor.

2. No Orthogonal Validation Performed

There is still no orthogonal validation (e.g., immunocytochemistry) of the key findings. The authors suggest this could be explored in future work. Validating altered pathways through straightforward immunocytochemical approaches should be part of the present analysis, not deferred.

3. fig 1D Unclear Representation of Western Blot Data

Regarding Figure 1d, the authors assert that three controls were used and Western blots were performed in triplicate. Yet only three data points are shown, raising doubts about whether all replicates are properly represented. The explanation provided fails to clarify this issue, especially given the substantial variation evident on the blots. To address this properly, the authors should display the individual replicates for each line and report statistics per line rather than presenting pooled averages. This is a basic expectation for transparent and interpretable data reporting.

**Have all data underlying the figures and results presented in the manuscript been provided?**

Reviewer #1: Yes

Reviewer #3: Yes

PLOS authors have the option to publish the peer review history of their article (what does this mean? ). If published, this will include your full peer review and any attached files.

**Do you want your identity to be public for this peer review?** For information about this choice, including consent withdrawal, please see our Privacy Policy .

Reviewer #1: No

Reviewer #3: **Yes: ** Nael Nadif Kasri

**Figure resubmission:**
---

## [Editor Report · Decision Letter 2]

2 Sep 2025

Dear Dr Martirosian,

We are pleased to inform you that your manuscript entitled "Epigenome and transcriptome changes in KMT2D-related Kabuki syndrome Type 1 iPSCs, neuronal progenitors and cortical neurons" has been editorially accepted for publication in PLOS Genetics. Congratulations!

Yours sincerely,

Gregory M. Cooper, PhD

Section Editor

PLOS Genetics

Gregory Cooper

Section Editor

PLOS Genetics

Aimée Dudley

Editor-in-Chief

PLOS Genetics

Anne Goriely

Editor-in-Chief

PLOS Genetics

Comments from the reviewers (if applicable):

**Data Deposition**

http://datadryad.org/submit?journalID=pgenetics&manu=PGENETICS-D-25-00138R2

**Press Queries**

---

## [Editor Report · Acceptance letter]

PGENETICS-D-25-00138R2

Epigenome and transcriptome changes in KMT2D-related Kabuki syndrome Type 1 iPSCs, neuronal progenitors and cortical neurons

Dear Dr Martirosian,

We are pleased to inform you that your manuscript entitled " 

Epigenome and transcriptome changes in KMT2D-related Kabuki syndrome Type 1 iPSCs, neuronal progenitors and cortical neurons" has been formally accepted for publication in PLOS Genetics! Your manuscript is now with our production department and you will be notified of the publication date in due course.

With kind regards,

Boglarka Nagy

PLOS Genetics

On behalf of:
